# Grounding Representation Similarity with Statistical Testing

**Frances Ding,  Jean-Stanislas Denain,  Jacob Steinhardt**
University of California Berkeley
{frances, js_denain, jsteinhardt}@berkeley.edu

## Abstract

To understand neural network behavior, recent works quantitatively compare different networks' learned representations using canonical correlation analysis (CCA), centered kernel alignment (CKA), and other dissimilarity measures. Unfortunately, these widely used measures often disagree on fundamental observations, such as whether deep networks differing only in random initialization learn similar representations. These disagreements raise the question: which, if any, of these dissimilarity measures should we believe? We provide a framework to ground this question through a concrete test: measures should have *sensitivity* to changes that affect functional behavior, and *specificity* against changes that do not. We quantify this through a variety of functional behaviors including probing accuracy and robustness to distribution shift, and examine changes such as varying random initialization and deleting principal components. We find that current metrics exhibit different weaknesses, note that a classical baseline performs surprisingly well, and highlight settings where all metrics appear to fail, thus providing a challenge set for further improvement.

## 1 Introduction

Understanding neural networks is not only scientifically interesting, but critical for applying deep networks in high-stakes situations. Recent work has highlighted the value of analyzing not just the final outputs of a network, but also its intermediate representations [20, 29]. This has motivated the development of representation similarity measures, which can provide insight into how different training schemes, architectures, and datasets affect networks' learned representations.

A number of similarity measures have been proposed, including centered kernel alignment (CKA) [13], ones based on canonical correlation analysis (CCA) [24, 30], single neuron alignment [20], vector space alignment [3, 6, 32], and others [2, 9, 16, 18, 21, 39]. Unfortunately, these different measures tell different stories. For instance, CKA and projection weighted CCA disagree on which layers of different networks are most similar [13]. This lack of consensus is worrying, as measures are often designed according to different and incompatible intuitive desiderata, such as whether finding a one-to-one assignment, or finding few-to-one mappings, between neurons is more appropriate [20]. As a community, we need well-chosen formal criteria for evaluating metrics to avoid over-reliance on intuition and the pitfalls of too many researcher degrees of freedom [17].

In this paper we view representation dissimilarity measures as implicitly answering a classification question–whether two representations are essentially similar or importantly different. Thus, in analogy to statistical testing, we can evaluate them based on their *sensitivity* to important change and *specificity* (non-responsiveness) against unimportant changes or noise.

As a warm-up, we first initially consider two intuitive criteria: first, that metrics should have specificity against random initialization; and second, that they should be sensitive to deleting important principal

components (those that affect probing accuracy). Unfortunately, popular metrics fail at least one of these two tests. CCA is not specific – random initialization noise overwhelms differences between even far-apart layers in a network (Section 3.1). CKA on the other hand is not sensitive, failing to detect changes in all but the top 10 principal components of a representation (Section 3.2).

We next construct quantitative benchmarks to evaluate a dissimilarity measure's quality. To move beyond our intuitive criteria, we need a ground truth. For this we turn to the functional behavior of the representations we are comparing, measured through probing accuracy (an indicator of syntactic information) [4, 27, 35] and out-of-distribution performance of the model they belong to [7, 23, 25]. We then score dissimilarity measures based on their rank correlation with these measured functional differences. Overall our benchmarks contain 30,480 examples and vary representations across several axes including random seed, layer depth, and low-rank approximation (Section 4)[1].

Our benchmarks confirm our two intuitive observations: on subtasks that consider layer depth and principal component deletion, we measure the rank correlation with probing accuracy and find CCA and CKA lacking as the previous warm-up experiments suggested. Meanwhile, the Orthogonal Procrustes distance, a classical but often overlooked[2] dissimilarity measure, balances gracefully between CKA and CCA and consistently performs well. This underscores the need for systematic evaluation, otherwise we may fall to recency bias that undervalues classical baselines.

Other subtasks measure correlation with OOD accuracy, motivated by the observation that random initialization sometimes has large effects on OOD performance [23]. We find that dissimilarity measures can sometimes predict OOD performance using only the in-distribution representations, but we also identify a challenge set on which none of the measures do statistically better than chance. We hope this challenge set will help measure and spur progress in the future.

## 2   Problem Setup: Metrics and Models

Our goal is to quantify the similarity between two different groups of neurons (usually layers). We do this by comparing how their activations behave on the same dataset. Thus for a layer with $p_1$ neurons, we define $A \in \mathbb{R}^{p_1 \times n}$, the matrix of activations of the $p_1$ neurons on $n$ data points, to be that layer's raw representation of the data. Similarly, let $B \in \mathbb{R}^{p_2 \times n}$ be a matrix of the activations of $p_2$ neurons on the same $n$ data points. We center and normalize these representations before computing dissimilarity, per standard practice. Specifically, for a raw representation $A$ we first subtract the mean value from each column, then divide by the Frobenius norm, to produce the normalized representation $A^*$, used in all our dissimilarity computations. In this work we study dissimilarity measures $d(A^*, B^*)$ that allow for quantitative comparisons of representations both within and across different networks. We colloquially refer to values of $d(A^*, B^*)$ as distances, although they do not necessarily satisfy the triangle inequality required of a proper metric.

We study five dissimilarity measures: centered kernel alignment (CKA), three measures derived from canonical correlation analysis (CCA), and a measure derived from the orthogonal Procrustes problem.

**Centered kernel alignment (CKA)** uses an inner product to quantify similarity between two representations. It is based on the idea that one can first choose a kernel, compute the $n \times n$ kernel matrix for each representation, and then measure similarity as the alignment between these two kernel matrices. The measure of similarity thus depends on one's choice of kernel; in this work we consider **Linear CKA**:

$$d_{\text{Linear CKA}}(A, B) = 1 - \frac{\|AB^\top\|_F^2}{\|AA^\top\|_F \|BB^\top\|_F} \tag{1}$$

as proposed in Kornblith et al. [13]. Other choices of kernel are also valid; we focus on Linear CKA here since Kornblith et al. [13] report similar results from using either a linear or RBF kernel.

**Canonical correlation analysis (CCA)** finds orthogonal bases $(w_A^i, w_B^i)$ for two matrices such that after projection onto $w_A^i, w_B^i$, the projected matrices have maximally correlated rows. For $1 \le i \le p_1$,

---

[1]Code to replicate our results can be found at `https://github.com/js-d/sim_metric`.
[2]For instance, Raghu et al. [30] and Morcos et al. [24] do not mention it, and Kornblith et al. [13] relegates it to the appendix; although Smith et al. [32] does use it to analyze word embeddings and prefers it to CCA.

the $i^{\text{th}}$ canonical correlation coefficient $\rho_i$ is computed as follows:

$$\rho_i = \max_{w_A^i, w_B^i} \frac{\langle w_A^{i\top} A, w_B^{i\top} B \rangle}{\|w_A^{i\top} A\| \cdot \|w_B^{i\top} B\|} \tag{2}$$

$$s.t. \ \langle w_A^{i\top} A, w_A^{j\top} A \rangle = 0, \ \forall j < i, \quad \langle w_B^{i\top} B, w_B^{j\top} B \rangle = 0, \ \forall j < i \tag{3}$$

To transform the vector of correlation coefficients into a scalar measure, two options considered previously [13] are the **mean correlation coefficient, $\bar{\rho}_{\text{CCA}}$**, and the **mean squared correlation coefficient, $R^2_{\text{CCA}}$**, defined as follows:

$$d_{\bar{\rho}_{\text{CCA}}}(A, B) = 1 - \frac{1}{p_1} \sum_i \rho_i, \qquad d_{R^2_{\text{CCA}}}(A, B) = 1 - \frac{1}{p_1} \sum_i \rho_i^2 \tag{4}$$

To improve the robustness of CCA, Morcos et al. [24] propose **projection-weighted CCA (PWCCA)** as another scalar summary of CCA:

$$d_{\text{PWCCA}}(A, B) = 1 - \frac{\sum_i \alpha_i \rho_i}{\sum_i \alpha_i}, \quad \alpha_i = \sum_j |\langle h_i, a_j \rangle| \tag{5}$$

where $a_j$ is the $j^{\text{th}}$ row of $A$, and $h_i = w_A^{i\top} A$ is the projection of $A$ onto the $i^{\text{th}}$ canonical direction. We find that PWCCA performs far better than $\bar{\rho}_{\text{CCA}}$ and $R^2_{\text{CCA}}$, so we focus on PWCCA in the main text, but include results on the other two measures in the appendix.

The **orthogonal Procrustes** problem consists of finding the left-rotation of $A$ that is closest to $B$ in Frobenius norm, i.e. solving the optimization problem:

$$\min_R \|B - RA\|_F^2, \quad \text{subject to } R^\top R = I. \tag{6}$$

The minimum is the squared **orthogonal Procrustes distance** between $A$ and $B$, and is equal to

$$d_{\text{Proc}}(A, B) = \|A\|_F^2 + \|B\|_F^2 - 2\|A^\top B\|_*, \tag{7}$$

where $\|\cdot\|_*$ is the nuclear norm [31]. Unlike the other metrics, the orthogonal Procrustes distance is not normalized between 0 and 1, although for normalized $A^*$, $B^*$ it lies in $[0, 2]$.

## 2.1 Models we study

In this work we study representations of both text and image inputs. For text, we investigate representations computed by Transformer architectures in the BERT model family [8] on sentences from the Multigenre Natural Language Inference (MNLI) dataset [40]. We study BERT models of two sizes: BERT base, with 12 hidden layers of 768 neurons, and BERT medium, with 8 hidden layers of 512 neurons. We use the same architectures as in the open source BERT release[3], but to generate diversity we study 3 variations of these models:

1. 10 BERT base models pretrained with different random seeds but not finetuned for particular tasks, released by Zhong et al. [41][4].
2. 10 BERT medium models initialized from pretrained models released by Zhong et al. [41], that we further finetuned on MNLI with 10 different finetuning seeds (100 models total).
3. 100 BERT base models that were initialized from the pretrained BERT model in [8] and finetuned on MNLI with different seeds, released by McCoy et al. [23][5].

For images, we investigate representations computed by ResNets [11] on CIFAR-10 test set images [14]. We train 100 ResNet-14 models[6] from random initialization with different seeds on the CIFAR-10 training set and collect representations after each convolutional layer.

Further training details, as well as checks that our training protocols result in models with comparable performance to the original model releases, can be found in Appendix A.

---

[3]available at https://github.com/google-research/bert
[4]available at https://github.com/ruiqi-zhong/acl2021-instance-level
[5]available at https://github.com/tommccoy1/hans/tree/master/berts_of_a_feather
[6]from https://github.com/pytorch/vision/blob/master/torchvision/models/resnet.py

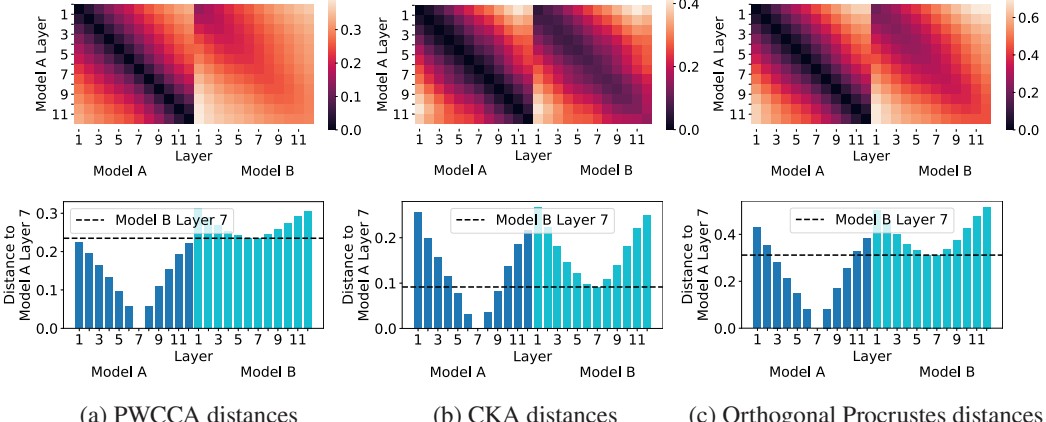

<table>
<tr><td>(a) PWCCA distances</td><td>(b) CKA distances</td><td>(c) Orthogonal Procrustes distances</td></tr>
</table>

Figure 1: **PWCCA fails the intuitive specificity test.** Top: PWCCA, CKA, and Orthogonal Procrustes pairwise distances between each layer of two differently initialized networks (Model A and B). Bottom: We zoom in to analyze the $7^{th}$ layer of Model A, plotting this layer's distance to every other layer in both networks; the dashed line indicates the distance to the corresponding $7^{th}$ layer in Model B. For PWCCA, none of the distances in model A exceed this line, indicating that random initialization affects this distance more than large changes in layer depth.

## 3 Warm-up: Intuitive Tests for Sensitivity and Specificity

When designing dissimilarity measures, researchers usually consider invariants that these measures should not be sensitive to [13]; for example, symmetries in neural networks imply that permuting the neurons in a fully connected layer does not change the representations learned. We take this one step further and frame dissimilarity measures as answering whether representations are essentially the same, or importantly different. We can then evaluate measures based on whether they respond to important changes (sensitivity) while ignoring changes that don't matter (specificity).

Assessing sensitivity and specificity requires a ground truth–which representations are truly different? To answer this, we begin with the following two intuitions[7]: 1) neural network representations trained on the same data but from different random initializations are similar, and 2) representations lose crucial information as principal components are deleted. These motivate the following intuitive tests of specificity and sensitivity: we expect a dissimilarity measure to: 1) assign a small distance between architecturally identical neural networks that only differ in initialization seed, and 2) assign a large distance between a representation $A$ and the representation $\hat{A}$ after deleting important principal components (enough to affect accuracy). We will see that PWCCA fails the first test (specificity), while CKA fails the second (sensitivity).

### 3.1 Specificity against changes to random seed

Neural networks with the same architecture trained from different random initializations show many similarities, such as highly correlated predictions on in-distribution data points [23]. Thus it seems natural to expect a good similarity measure to assign small distances between architecturally corresponding layers of networks that are identical except for initialization seed.

To check this property, we take two BERT base models pre-trained with different random seeds and, for every layer in the first model, compute its dissimilarity to every layer in both the first and second model. We do this for 5 separate pairs of models and average the results. To pass the intuitive specificity test, a dissimilarity measure should assign relatively small distances between a layer in the first network and its corresponding layer in the second network.

Figure 1 displays the average pair-wise PWCCA, CKA, and Orthogonal Procrustes distances between layers of two networks differing only in random seed. According to PWCCA, these networks' representations are quite dissimilar; for instance, the two layer 7 representations are further apart

---

[7]Note we will see later that these intuitions need refinement.

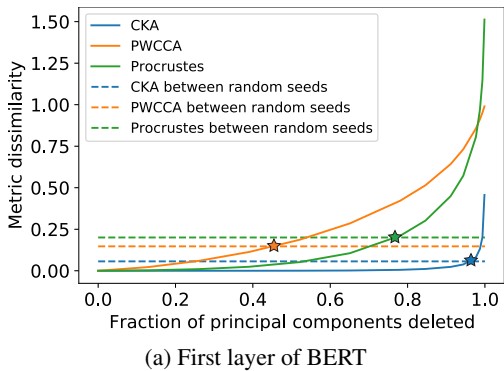

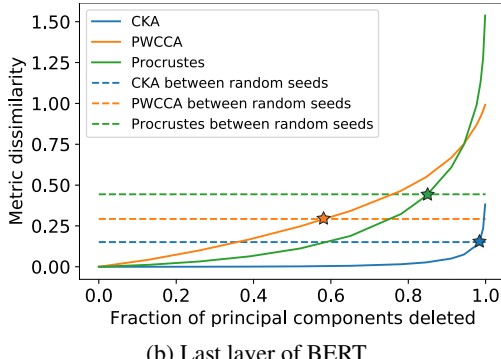

(a) First layer of BERT                          (b) Last layer of BERT

Figure 2: **CKA fails to be sensitive to all but the largest principal components.** We compute dissimilarities between a layer's representation and low-rank approximations to that representation obtained by deleting principal components, starting from the smallest (solid lines). We also compute the average distance between networks trained with different random seeds as a baseline (dotted line), and mark the intersection point with a star. The starred points indicate that CKA requires almost all the components to be deleted before CKA distance exceeds the baseline.

than they are from any other layer in the same network. PWCCA is thus not specific against random initialization, as it can outweigh even large changes in layer depth.

In contrast, CKA can separate layer 7 in a different network from layers 4 or 10 in the same network, showing better specificity to random initialization. Orthogonal Procrustes exhibits smaller but non-trivial specificity, distinguishing layers once they are 4-5 layers apart.

## 3.2 Sensitivity to removing principal components

Dissimilarity measures should also be sensitive to deleting important principal components of a representation.[8] To quantify which components are important, we fix a layer of a pre-trained BERT base model and measure how probing accuracy degrades as principal components are deleted (starting from the smallest component), since probing accuracy is a common measure of the information captured in a representation [4]. We probe linear classification performance on the Stanford Sentiment Tree Bank task (SST-2) [33], following the experimental protocol in Tamkin et al. [34]. Figure 3b shows how probing accuracy degrades with component deletion. Ideally, dissimilarity measures should be large by the time probing accuracy has decreased substantially.

To assess whether a dissimilarity measure is large, we need a baseline to compare to. For each measure, we define a dissimilarity score to be above the *detectable* threshold if it is larger than the dissimilarity score between networks with different random initialization. Figure 2 plots the dissimilarity induced by deleting principal components, as well as this baseline.

For the last layer of BERT, CKA requires 97% of a representation's principal components to be deleted for the dissimilarity to be detectable; after deleting these components, probing accuracy shown in Figure 3b drops significantly from 80% to 63% (chance is 50%). CKA thus fails to detect large accuracy drops and so fails our intuitive sensitivity test.

Other metrics perform better: Orthogonal Procrustes's detection threshold is ∼85% of the principal components, corresponding to an accuracy drop 80% to 70%. PWCCA's threshold is ∼55% of principal components, corresponding to an accuracy drop from 80% to 75%.

PWCCA's failure of specificity and CKA's failure of sensitivity on these intuitive tests are worrying. However, before declaring definitive failure, in the next section, we turn to making our assessments more rigorous.

---

[8]For a representation $A$, we define $\hat{A}_{-k}$, the result of deleting the $k$ smallest principal components from $A$, as follows: we compute the singular value decomposition $U\Sigma V^T = A$, construct $U_{-k} \in \mathbb{R}^{p \times p-k}$ by dropping the lowest $k$ singular vectors of $U$, and finally take $\hat{A}_{-k} = U_{-k}^T A$.

## 4 Rigorously Evaluating Dissimilarity Metrics

In the previous section, we saw that CKA and PWCCA each failed intuitive tests, based on sensitivity to principal components and specificity to random initialization. However, these were based primarily on intuitive, qualitative desiderata. Is there some way for us to make these tests more rigorous and quantitative?

First consider the intuitive layer specificity test (Section 3.1), which revealed that random initialization affects PWCCA more than large changes in layer depth. To justify why this is undesirable, we can turn to probing accuracy, which is strongly affected by layer depth, and only weakly affected by random seed (Figure 3a). This suggests a path forward: we can ground the layer test in the concrete differences in functionality captured by the probe.

More generally, we want metrics to be sensitive to changes that affect functionality, while ignoring those that don't. This motivates the following general procedure, given a distance metric $d$ and a functionality $f$ (which assigns a real number to a given representation):

1. Collect a set $S$ of representations that differ along one or more axes of interest (e.g. layer depth, random seed).
2. Choose a reference representation $A \in S$. When $f$ is an accuracy metric, it is reasonable to choose $A = \arg\max_{A \in S} f(A)$.[9]
3. For every representation $B \in S$:
   - Compute $|f(A) - f(B)|$
   - Compute $d(A, B)$
4. Report the rank correlation between $|f(A) - f(B)|$ and $d(A, B)$ (measured by Kendall's $\tau$ or Spearman $\rho$).

The above procedure provides a *quantitative* measure of how well the distance metric $d$ responds to the functionality $f$. For instance, in the layer specificity test, since depth affects probing accuracy strongly while random seed affects it only weakly, a dissimilarity measure with high rank correlation will be strongly responsive to layer depth and weakly responsive to seed; thus rank correlation quantitatively formalizes the test from Section 3.1.

Correlation metrics also capture properties that our intuition might miss. For instance, Figure 3a shows that some variation in random seed actually does affect accuracy, and our procedure rewards metrics that pick up on this, while the intuitive sensitivity test would penalize them.

Our procedure requires choosing a collection of models $S$; the crucial feature of $S$ is that it contains models with diverse behavior according to $f$. Different sets $S$, combined with a functional difference $f$, can be thought of as miniature "benchmarks" that surface complementary perspectives on dissimilarity measures' responsiveness to that functional difference. In the rest of this section, we instantiate this quantitative benchmark for several choices of $f$ and $S$, starting with the layer and principal component tests from Section 3 and continuing on to several tests of OOD performance.

The overall results are summarized in Table 1. Note that for any single benchmark, we expect the correlation coefficients to be significantly lower than 1, since the metric $D$ must capture all important axes of variation while $f$ measures only one type of functionality. A good metric is one that has consistently high correlation across many different functional measures.

**Benchmark 1: Layer depth.** We turn the layer test into a benchmark for both text and images. For the text setting, we construct a set $S$ of 120 representations by pretraining 10 BERT base models with different initialization seeds and including each of the 12 BERT layers as a representation. We separately consider two functionalities $f$: probing accuracy on QNLI [37] and SST-2 [33]. To compute the rank correlation, we take the reference representation $A$ to be the representation with highest probing accuracy. We compute the Kendall's $\tau$ and Spearman's $\rho$ rank correlations between the dissimilarities and the probing accuracy differences and report the results in Table 1.

---

[9]Choosing the highest accuracy model as the reference makes it more likely that as accuracy changes, models are on average becoming more dissimilar. A low accuracy model may be on the "periphery" of model space, where it is dissimilar to models with high accuracy, but potentially even more dissimilar to other low accuracy models that make different mistakes.

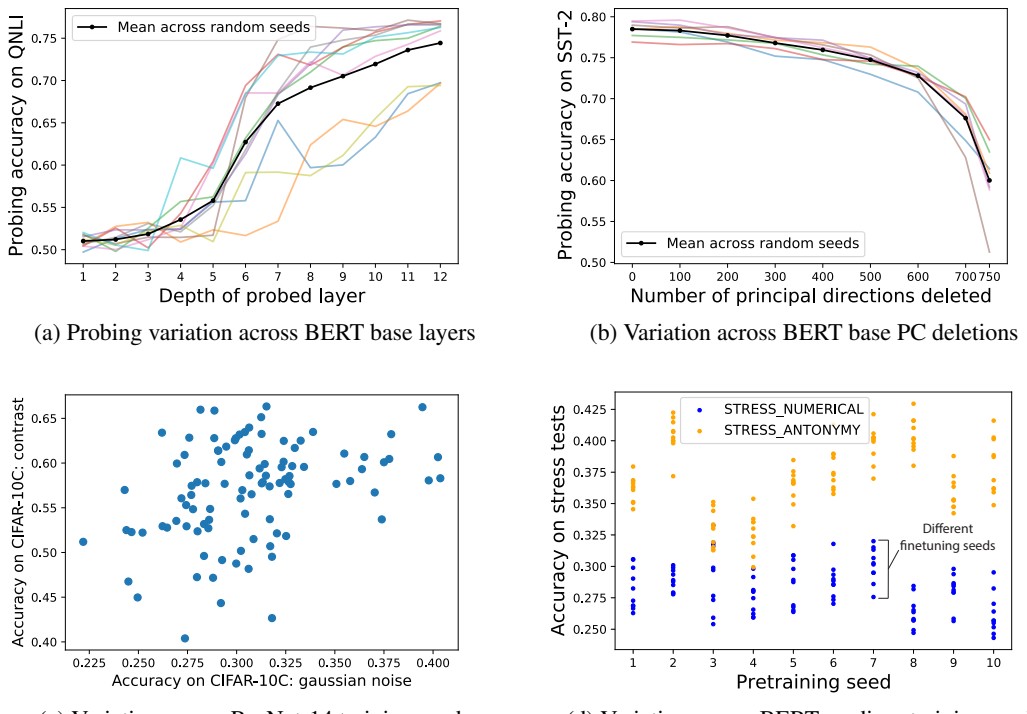

(a) Probing variation across BERT base layers

(b) Variation across BERT base PC deletions

(c) Variation across ResNet-14 training seeds

(d) Variation across BERT medium training seeds

Figure 3: **Our perturbations induce substantial variation on probing tasks and stress tests:** (3a): Changing the depth of the examined BERT base layer strongly affects probing accuracy on QNLI. The trend for each randomly initialized model is displayed semi-transparently, and the solid black line is the mean trend. (3b): Truncating principal components from pretrained BERT base significantly degrades probing accuracy on SST-2 (BERT layer 12 shown here). (3c): Training ResNet-14 on CIFAR-10 with different seeds leads to variation in accuracies on CIFAR-10C corruptions (here Gaussian noise and contrast). (3d): Pretraining and finetuning BERT medium with 10 different pretraining seeds and 10 different finetuning seeds per pretrained model leads to variation in accuracies on the Antonymy (yellow scatter points) and Numerical (blue scatter points) stress tests [25].

For the image setting, we similarly construct a set $S$ of 70 representations by training 5 ResNet-14 models with different initialization seeds and including each of the 14 layers' representations. We also consider two functionalities $f$ for these vision models: probing accuracy on CIFAR-100 [14] and on SVHN [26], and compute rank correlations in the same way.

We find that PWCCA has lower rank correlations compared to CKA and Procrustes for both language probing tasks. This corroborates the intuitive specificity test (Section 3.1), suggesting that PWCCA registers too large of a dissimilarity across random initializations. For the vision tasks, CKA and Procrustes achieve similar rank correlations, while PWCCA cannot be computed because $n < d$.

**Benchmark 2: Principal component (PC) deletion.** We next quantify the PC deletion test from Section 3.2, by constructing a set $S$ of representations that vary in both random initialization and fraction of principal components deleted. We pretrain 10 BERT base models with different initializations, and for each pretrained model we obtain 14 different representations by deleting that representation's $k$ smallest principal components, with $k \in \{0, 100, 200, 300, 400, 500, 600, 650, 700, 725, 750, 758, 763, 767\}$. Thus $S$ has $10 \times 14 = 140$ elements. The representations themselves are the layer-$\ell$ activations, for $\ell \in \{8, 9, \dots, 12\}$,[10] so there are 5 different choices of $S$. We use SST-2 probing accuracy as the functionality of interest $f$, and select the reference representation $A$ as the element in $S$ with highest accuracy. Rank correlation

---

[10]Earlier layers have near-chance accuracy on probing tasks, so we ignore them.

Table 1: **Summary of rank correlation results.** For Benchmarks #1-3 in both language and vision, all dissimilarity measures successfully achieve significant positive rank correlation with the functionality of interest–both CKA and PWCCA dominate certain benchmarks and fall behind on others, while Procrustes is more consistent and often close to the leader. Benchmark #4 is more challenging, and no dissimilarity measure achieves a high correlation. The vision experiments do not have results for PWCCA because $n < d$.

| # | Perturbation | Subtask Size | Functionality $\rho$ | Procrustes $\tau$ | Procrustes $\rho$ | CKA $\tau$ | CKA $\rho$ | PWCCA $\tau$ | PWCCA $\rho$ |
|---|---|---|---|---|---|---|---|---|---|
| | | | Modality: Language | | | | | | |
| 1 | Pretraining seed, layer depth | 120 | Probe: QNLI | 0.862 | 0.670 | **0.876** | **0.685** | 0.763 | 0.564 |
| | | 120 | Probe: SST-2 | 0.890 | 0.707 | **0.905** | **0.732** | 0.829 | 0.637 |
| 2 | Pretraining seed, PC deletion | 140 × 5 | Probe: SST-2 | 0.860 | 0.677 | 0.751 | 0.564 | **0.870** | **0.690** |
| 3 | Finetuning seed | 100 × 12 | OOD: HANS Lexical non-entailed | 0.551 | 0.398 | 0.462 | 0.329 | **0.568** | **0.412** |
| 4 | Pretraining and finetuning seeds | 100 × 8 | OOD: Antonymy stress test | **0.243** | **0.178** | 0.227 | 0.160 | 0.204 | 0.152 |
| | | 100 × 8 | OOD: Numerical stress test | 0.071 | 0.049 | **0.122** | **0.084** | 0.031 | 0.023 |
| | Total (language) | 3740 | Average | **0.580** | **0.447** | 0.557 | 0.426 | 0.544 | 0.413 |
| | | | Modality: Vision | | | | | | |
| 1 | Training seed, layer depth | 70 | Probe: CIFAR-100 | 0.485 | **0.376** | **0.507** | 0.359 | - | - |
| | | 70 | Probe: SVHN | 0.363 | **0.272** | **0.372** | 0.255 | - | - |
| 4 | Training seed | 1900 × 14 | OOD: CIFAR-10C | **0.060** | **0.057** | 0.041 | 0.038 | - | - |
| | Total (vision) | 26740 | Average | 0.303 | **0.235** | **0.307** | 0.217 | - | - |

results are consistent across the 5 choices of $S$ (Appendix C), so we report the average as a summary statistic in Table 1.

We find that PWCCA has the highest rank correlation between dissimilarity and probing accuracy, followed by Procrustes, and distantly followed by CKA. This corroborates the intuitive observations from Section 3.2 that CKA is not sensitive to principal component deletion.

### 4.1 Investigating variation in OOD performance across random seeds

So far our benchmarks have been based on probing accuracy, which only measures in-distribution behavior (the train and test set of the probe are typically i.i.d.). In addition, the BERT models were always pretrained on language modeling but not finetuned for classification. To add diversity to our benchmarks, we next consider the out-of-distribution performance of language and vision models trained for classification tasks.

**Benchmark 3: Changing fine-tuning seeds.** McCoy et al. [23] show that a single pretrained BERT base model finetuned on MNLI with different random initializations will produce models with similar in-distribution performance, but widely variable performance on out-of-distribution data. We thus create a benchmark $S$ out of McCoy et al.'s 100 released fine-tuned models, using OOD accuracy on the "Lexical Heuristic (Non-entailment)" subset of the HANS dataset [22] as our functionality $f$. This functionality is associated with the entire model, rather than an individual layer (in contrast to the probing functionality), but we consider one layer at a time to measure whether dissimilarities

between representations at that layer correlate with $f$. This allows us to also localize whether certain layers are more predictive of $f$.

We construct 12 different $S$ (one for each of the 12 layers of BERT base), taking the reference representation $A$ to be that of the highest accuracy model according to $f$. As before, we report each dissimilarity measure's rank correlation with $f$ in Table 1, averaged over the 12 runs.

All three dissimilarity measures correlate with OOD accuracy, with Orthogonal Procrustes and PWCCA being more correlated than CKA. Since the representations in our benchmarks were computed on in-distribution MNLI data, this has the interesting implication that dissimilarity measures can detect OOD differences without access to OOD data. It also implies that random initialization leads to meaningful functional differences that are picked up by these measures, especially Procrustes and PWCCA. Contrast this with our intuitive specificity test in Section 3.1, where all sensitivity to random initialization was seen as a shortcoming. Our more quantitative benchmark here suggests that some of that sensitivity tracks true functionality.

To check that the differences in rank correlation for Procrustes, PWCCA, and CKA are statistically significant, we compute bootstrap estimates of their 95% confidence intervals. With 2000 bootstrapped samples, we find statistically significant differences between all pairs of measures for most choices of layer depth $S$, so we conclude PWCCA > Orthogonal Procrustes > CKA (the full results are in Appendix E). We do not apply this procedure for the previous two benchmarks, because the different models have correlated randomness and so any $p$-value based on independence assumptions would be invalid.

**Benchmark 4: Challenge sets: Changing pretraining and fine-tuning seeds.**    We also construct benchmarks using models trained from scratch with different random seeds (for language, this is pretraining and fine-tuning, and for vision, this is standard training). For language, we construct benchmarks from a collection of 100 BERT medium models, trained with all combinations of 10 pretraining and 10 fine-tuning seeds. The models are fine-tuned on MNLI, and we consider two different functionalities of interest $f$: accuracy on the OOD Antonymy stress test and on the OOD Numerical stress test [25], which both show significant variation in accuracy across models (see Figure 3d). We obtain 8 different sets $S$ (one for each of the 8 layer depths in BERT medium), again taking $A$ to be the representation of the highest-accuracy model according to $f$. Rank correlations for each dissimilarity measure are averaged over the 8 runs and reported in Table 1.

For vision, we construct benchmarks from a collection of 100 ResNet-14 models, trained with different random seeds on CIFAR-10. We consider 19 different functionalities of interest—the 19 types of corruptions in the CIFAR-10C dataset [12], which show significant variation in accuracy across models (see Figure 3c). We obtain 14 different sets $S$ (one for each of the 14 layers), taking $A$ to be the representation of the highest-accuracy model according to $f$. Rank correlations for each dissimilarity measure are averaged over the 14 runs and over the 19 corruption types and reported in Table 1. Results for each of the 19 corruptions individually can be found in Appendix D..

None of the dissimilarity measures show a large rank correlation for either the language or vision tasks, and for the Numerical stress test, at most layers, the associated $p$-values (assuming independence) are non-significant at the 0.05 level (see Appendix C). [11] Thus we conclude that all measures fail to be sensitive to OOD accuracy in these settings. One reason for this could be that there is less variation in the OOD accuracies compared to the previous experiment with the HANS dataset (there accuracies varied from 0 to nearly 60%). Another reason could be that it is harder to correctly account for both pretraining and fine-tuning variation at the same time. Either way, we hope that future dissimilarity measures can improve upon these results, and we present this benchmark as a challenge task to motivate progress.

## 5   Discussion

In this work we proposed a quantitative measure for evaluating similarity metrics, based on the rank correlation with functional behavior. Using this, we generated tasks motivated by sensitivity to

---

[11] See Appendix C for $p$-values as produced by sci-kit learn. Strictly speaking, the $p$-values are invalid because they assume independence, but the pretraining seed induces correlations. However, correctly accounting for these would tend to make the $p$-values larger, thus preserving our conclusion of non-significance .

deleting important directions, specificity to random initialization, and sensitivity to out-of-distribution performance. Popular existing metrics such as CKA and CCA often performed poorly on these tasks, sometimes in striking ways. Meanwhile, the classical Orthogonal Procrustes transform attained consistently good performance.

Given the success of Orthogonal Procrustes, it is worth reflecting on how it differs from the other metrics and why it might perform well. To do so, we consider a simplified case where $A$ and $B$ have the same singular vectors but different singular values. Thus without loss of generality $A = \Lambda_1$ and $B = \Lambda_2$, where the $\Lambda_i$ are both diagonal. In this case, the Orthogonal Procrustes distance reduces to $\|\Lambda_1 - \Lambda_2\|_F^2$, or the sum of the squared distances between the singular values. We will see that both CCA and CKA reduce to less reasonable formulae in this case.

*Orthogonal Procrustes vs. CCA.* All three metrics derived from CCA assign *zero* distance even when the (non-zero) singular values are arbitrarily different. This is because CCA correlation coefficients are invariant to all invertible linear transformations. This invariance property may help explain why CCA metrics generally find layers within the same network to be much more similar than networks trained with different randomness. Random initialization introduces noise, particularly in unimportant principal components, while representations within the same network more easily preserve these components, and CCA may place too much weight on their associated correlation coefficients.

*Orthogonal Procrustes vs. CKA.* In contrast to the squared distance of Orthogonal Procrustes, CKA actually reduces to a quartic function based on the dot products between the *squared* entries of $\Lambda_1$ and $\Lambda_2$. As a consequence, CKA is dominated by representations' largest singular values, leaving it insensitive to meaningful differences in smaller singular values as illustrated in Figure 2. This lack of sensitivity to moderate-sized differences may help explain why CKA fails to track out-of-distribution error effectively.

In addition to helping understand similarity measures, our benchmarks pinpoint directions for improvement. No method was sensitive to accuracy on the Numerical stress test in our challenge set, possibly due to a lower signal-to-noise ratio. Since Orthogonal Procrustes performed well on most of our tasks, it could be a promising foundation for a new measure, and recent work shows how to regularize Orthogonal Procrustes to handle high noise [28]. Perhaps similar techniques could be adapted here.

An alternative to our benchmarking approach is to directly define two representations' dissimilarity as their difference in a functional behavior of interest. Feng et al. [9] take this approach, defining dissimilarity as difference in accuracy on a handful of probing tasks. One drawback of this approach is that a small set of probes may not capture all the differences in representations, so it is useful to base dissimilarity measures on representations' intrinsic properties. Intrinsically defined dissimilarities also have the potential to highlight new functional behaviors, as we found that representations with similar in-distribution probing accuracy often have highly variable OOD accuracy.

A limitation of our work is that we only consider a handful of model variations and functional behaviors, and restricting our attention to these settings could overlook other important considerations. To address this, we envision a paradigm in which a rich tapestry of benchmarks are used to ground and validate neural network interpretations. Other axes of variation in models could include training on more or fewer examples, training on shuffled labels vs. real labels, training from specifically chosen initializations [10], and using different architectures. Other functional behaviors to examine could include modularity and meta-learning capabilities. Benchmarks could also be applied to other interpretability tools beyond dissimilarity. For example, sensitivity to deleting principal components could provide an additional sanity check for saliency maps and other visualization tools [1].

More broadly, many interpretability tools are designed as *audits* of models, although it is often unclear what characteristics of the models are consistently audited. We position this work as a *counter-audit*, where by collecting models that differ in functional behavior, we can assess whether the interpretability tools CKA, PWCCA, etc., accurately reflect the behavioral differences. Many other types of counter-audits may be designed to assess other interpretability tools. For example, models that have backdoors built into them to misclassify certain inputs provide counter-audits for interpretability tools that explain model predictions–these explanations should reflect any backdoors present [5, 15, 19, 38]. We are hopeful that more comprehensive checks on interpretability tools will provide deeper understanding of neural networks, and more reliable models.

## Acknowledgments and Disclosure of Funding

Thanks to Ruiqi Zhong for helpful comments and assistance in finetuning models, and thanks to Daniel Rothchild and our anonymous reviewers for helpful discussion. FD is supported by the National Science Foundation Graduate Research Fellowship Program under Grant No. DGE 1752814 and the Open Philanthropy Project AI Fellows Program. JSD is supported by the NSF Division of Mathematical Sciences Grant No. 2031985.

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
