# Appendix

## A  Training details

### A.1  BERT finetuning details

We fine-tuned models from Zhong et al. [41] and the original BERT models from Devlin et al. [8] on three tasks – Quora Question Pairs (QQP)[12], Multi-Genre Natural Language Inference (MNLI; Williams et al. [40]), and the Stanford Sentiment Treebank (SST-2; Socher et al. [33]), and show each model's accuracy on these tasks in Table 2. Our models generally have comparable accuracy.

As in Turc et al. [36], we finetune for 4 epochs for each dataset. For each task and model size, we tune hyperparameters in the following way: we first randomly split our new training set into 80% and 20%; then we finetune on the 80% split with all 9 combination of batch size [16, 32, 64] and learning rate [1e-4, 5e-5, 3e-5], and choose the combination that leads to the best average accuracy on the remaining 20%. Finetuning these models for all three tasks requires around 500 hours.

Table 2: Comparing accuracy of our pretrained model (superscript $^{ours}$) to the original release by Devlin et al. [8] and Turc et al. [36] (superscript $^{orig}$) on a variety of fine-tuned tasks.

|  | QQP | MNLI | SST-2 |
|---|---|---|---|
| BERT medium $^{orig}$ | 89.8% | 79.6% | 94.2% |
| BERT medium $^{ours}$ | 89.5% | 78.9% | 94.2% |
| BERT base $^{orig}$ | 90.8% | 83.8% | 95.0% |
| BERT base $^{ours}$ | 90.6% | 81.2% | 94.6% |

### A.2  ResNet training details

We trained ResNet-14 models on CIFAR-10 training data with the following hyperparameters:

- learning rate: 0.1
- epochs: 100
- learning rate decay: 0.1 at epoch 50 and epoch 75
- batch size: 128

The 100 models we trained have an average accuracy on the CIFAR-10 test set of 90.2%, with standard deviation 0.2%. Training these models requires around 20 hours.

## B  Licenses

The source code for BERT models available at `https://github.com/google-research/bert` is licensed under the Apache License 2.0.

The model weights for the 100 BERT base models provided by McCoy et al. [23] are licensed under the Creative Commons Attribution 4.0 International license, and their source code is licensed under the MIT license (`https://github.com/tommccoy1/hans/blob/master/LICENSE.md`).

## C  Layer-wise results

Some of the results presented in Table 1 were averaged over multiple layers, since rankings between dissimilarity measures were consistent across different layers. Rank correlation scores are higher across all measures for certain layers, however, so we include layer-by-layer results here for completeness. We also include scores for $\bar{\rho}_{CCA}$ and $R^2_{CCA}$ here, and note that they are often similar to PWCCA, and generally dominated by other measures. We expand each row of Table 1 into a subsection of its own. We also include p-values as reported by sci-kit learn, although we note that because random

---

[12]https://www.quora.com/q/quoradata/First-Quora-Dataset-Release-Question-Pairs

seeds are shared among some representations, these p-values are all inflated, with the exception of those for the experiment perturbing only fine-tuning seed, and assessing functionality through HANS (C.3). The invalid p-values may all be thought of as upper-bounds for the significance of the rank correlation results.

### C.1 Perturbation: pretraining seed and layer depth

Tables 3 and 4 show the full results (including p-values and all 5 dissimilarity measures) using the QNLI probe as the functionality of interest, for Spearman $\rho$ and Kendall's $\tau$, respectively. Table 5 and 6 present results for the probing task SST-2 as the functionality of interest.

Table 3: Spearman $\rho$ results for perturbing pretraining seed and layer depth, and assessing functionality through the QNLI probe

| Layer | Procrustes | CKA | PWCCA | $\bar{\rho}_{\mathrm{CCA}}$ | $R^2_{\mathrm{CCA}}$ |
|---|---|---|---|---|---|
| 12 | 0.862 (6.5E-37) | 0.876 (1.6E-39) | 0.763 (2.2E-24) | 0.849 (1.0E+00) | 0.846 (1.0E+00) |

Table 4: Kendall's $\tau$ results for perturbing pretraining seed and layer depth, and assessing functionality through the QNLI probe

| Layer | Procrustes | CKA | PWCCA | $\bar{\rho}_{\mathrm{CCA}}$ | $R^2_{\mathrm{CCA}}$ |
|---|---|---|---|---|---|
| 12 | 0.670 (1.1E-27) | 0.685 (7.4E-29) | 0.564 (3.2E-20) | 0.652 (1.0E+00) | 0.647 (1.0E+00) |

Table 5: Spearman $\rho$ results for perturbing pretraining seed and layer depth, and assessing functionality through the SST-2 probe

| Layer | Procrustes | CKA | PWCCA | $\bar{\rho}_{\mathrm{CCA}}$ | $R^2_{\mathrm{CCA}}$ |
|---|---|---|---|---|---|
| 12 | 0.890 (2.7E-42) | 0.905 (5.3E-46) | 0.829 (7.7E-32) | 0.857 (1.0E+00) | 0.854 (1.0E+00) |

Table 6: Kendall's $\tau$ results for perturbing pretraining seed and layer depth, and assessing functionality through the SST-2 probe

| Layer | Procrustes | CKA | PWCCA | $\bar{\rho}_{\mathrm{CCA}}$ | $R^2_{\mathrm{CCA}}$ |
|---|---|---|---|---|---|
| 12 | 0.707 (1.2E-30) | 0.732 (1.0E-32) | 0.637 (3.1E-25) | 0.662 (1.0E+00) | 0.658 (1.0E+00) |

## C.2 Perturbation: pretraining seed and principal component deletion

We find that for these experiments, results are consistent across the layers we analyze (the last 6 layers of BERT base). Tables 7 and 8 show results for Spearman $\rho$ and Kendall's $\tau$, respectively.

Table 7: Layer-wise Spearman $\rho$ results for perturbing pretraining seed and principal component deletion, and assessing functionality through the SST-2 probe

| Layer | Procrustes | CKA | PWCCA | $\bar{\rho}_{\text{CCA}}$ | $R^2_{\text{CCA}}$ |
|-------|------------|-----|-------|---------------------------|--------------------|
| 8 | 0.764 (2.4E-36) | 0.668 (3.2E-25) | 0.776 (3.4E-38) | 0.700 (1.9E-28) | 0.700 (1.8E-28) |
| 9 | 0.813 (2.1E-44) | 0.706 (4.0E-29) | 0.825 (9.2E-47) | 0.728 (1.3E-31) | 0.728 (1.2E-31) |
| 10 | 0.873 (2.1E-58) | 0.818 (2.7E-45) | 0.874 (1.1E-58) | 0.748 (3.2E-34) | 0.749 (2.7E-34) |
| 11 | 0.918 (1.2E-74) | 0.797 (1.4E-41) | 0.922 (1.7E-76) | 0.781 (6.6E-39) | 0.781 (7.0E-39) |
| 12 | 0.932 (1.1E-81) | 0.766 (1.1E-36) | 0.955 (4.2E-97) | 0.810 (6.1E-44) | 0.810 (6.1E-44) |

Table 8: Layer-wise Kendall's $\tau$ results for perturbing pretraining seed and principal component deletion, and assessing functionality through the SST-2 probe

| Layer | Procrustes | CKA | PWCCA | $\bar{\rho}_{\text{CCA}}$ | $R^2_{\text{CCA}}$ |
|-------|------------|-----|-------|---------------------------|--------------------|
| 8 | 0.560 (1.8E-29) | 0.479 (4.4E-22) | 0.573 (1.1E-30) | 0.512 (6.8E-25) | 0.512 (6.6E-25) |
| 9 | 0.602 (1.2E-33) | 0.509 (1.2E-24) | 0.618 (2.5E-35) | 0.542 (1.1E-27) | 0.543 (9.7E-28) |
| 10 | 0.684 (5.6E-43) | 0.627 (2.1E-36) | 0.685 (5.3E-43) | 0.588 (2.9E-32) | 0.589 (2.5E-32) |
| 11 | 0.751 (2.8E-51) | 0.616 (3.3E-35) | 0.756 (6.4E-52) | 0.648 (9.2E-39) | 0.648 (9.2E-39) |
| 12 | 0.787 (3.4E-56) | 0.588 (2.9E-32) | 0.819 (1.2E-60) | 0.701 (4.7E-45) | 0.701 (4.9E-45) |

## C.3 Perturbation: fine-tuning seed, Functionality: HANS

Results for this experiment are similar across layers for Procrustes and all three CCA-based measures, with middle layers of BERT base having a slightly higher rank correlation score in general. For CKA, this effect is even more pronounced. Tables 9 and 10 show the results for Spearman $\rho$ and Kendall's $\tau$, respectively.

Table 9: Layer-wise Spearman $\rho$ results for perturbing finetuning seed, and assessing functionality through the HANS: Lexical (non-entailment) OOD dataset

| Layer | Procrustes ($p$) | CKA ($p$) | PWCCA ($p$) | $\bar{\rho}_{\text{CCA}}$ ($p$) | $R^2_{\text{CCA}}$ ($p$) |
|---|---|---|---|---|---|
| 1 | 0.425 (5.1E-06) | 0.361 (1.1E-04) | 0.405 (1.4E-05) | 0.388 (3.4E-05) | 0.389 (3.2E-05) |
| 2 | 0.510 (3.1E-08) | 0.410 (1.2E-05) | 0.486 (1.5E-07) | 0.488 (1.3E-07) | 0.483 (1.8E-07) |
| 3 | 0.531 (6.6E-09) | 0.427 (4.6E-06) | 0.538 (3.8E-09) | 0.533 (5.6E-09) | 0.532 (6.2E-09) |
| 4 | 0.543 (2.6E-09) | 0.506 (3.9E-08) | 0.552 (1.4E-09) | 0.555 (1.0E-09) | 0.550 (1.5E-09) |
| 5 | 0.563 (5.3E-10) | 0.512 (2.6E-08) | 0.570 (2.9E-10) | 0.582 (1.1E-10) | 0.580 (1.3E-10) |
| 6 | 0.629 (1.2E-12) | 0.641 (3.6E-13) | 0.621 (2.8E-12) | 0.621 (2.7E-12) | 0.622 (2.5E-12) |
| 7 | 0.647 (1.7E-13) | 0.658 (5.0E-14) | 0.647 (1.7E-13) | 0.653 (9.0E-14) | 0.650 (1.2E-13) |
| 8 | 0.643 (2.7E-13) | 0.552 (1.3E-09) | 0.653 (9.5E-14) | 0.651 (1.1E-13) | 0.651 (1.2E-13) |
| 9 | 0.589 (5.9E-11) | 0.419 (7.1E-06) | 0.641 (3.5E-13) | 0.662 (3.3E-14) | 0.660 (4.2E-14) |
| 10 | 0.536 (4.6E-09) | 0.437 (2.7E-06) | 0.559 (7.3E-10) | 0.612 (6.6E-12) | 0.614 (5.4E-12) |
| 11 | 0.532 (6.2E-09) | 0.426 (4.9E-06) | 0.565 (4.7E-10) | 0.619 (3.4E-12) | 0.614 (5.5E-12) |
| 12 | 0.465 (5.3E-07) | 0.192 (2.8E-02) | 0.574 (2.1E-10) | 0.609 (9.2E-12) | 0.610 (7.9E-12) |

Table 10: Layer-wise Kendall's $\tau$ results for perturbing finetuning seed, and assessing functionality through the HANS: Lexical (non-entailment) OOD dataset

| Layer | Procrustes ($p$) | CKA ($p$) | PWCCA ($p$) | $\bar{\rho}_{\text{CCA}}$ ($p$) | $R^2_{\text{CCA}}$ ($p$) |
|---|---|---|---|---|---|
| 1 | 0.295 (6.7E-06) | 0.269 (3.6E-05) | 0.277 (2.2E-05) | 0.265 (4.7E-05) | 0.268 (4.0E-05) |
| 2 | 0.363 (4.6E-08) | 0.288 (1.1E-05) | 0.343 (2.1E-07) | 0.342 (2.3E-07) | 0.342 (2.4E-07) |
| 3 | 0.372 (2.1E-08) | 0.290 (9.5E-06) | 0.378 (1.3E-08) | 0.375 (1.6E-08) | 0.375 (1.6E-08) |
| 4 | 0.393 (3.4E-09) | 0.358 (6.6E-08) | 0.401 (1.7E-09) | 0.405 (1.2E-09) | 0.403 (1.4E-09) |
| 5 | 0.410 (7.7E-10) | 0.367 (3.3E-08) | 0.417 (4.1E-10) | 0.428 (1.4E-10) | 0.424 (2.0E-10) |
| 6 | 0.464 (4.2E-12) | 0.474 (1.5E-12) | 0.460 (6.3E-12) | 0.460 (5.8E-12) | 0.461 (5.6E-12) |
| 7 | 0.483 (5.5E-13) | 0.488 (3.3E-13) | 0.481 (7.1E-13) | 0.486 (3.9E-13) | 0.483 (5.5E-13) |
| 8 | 0.478 (9.2E-13) | 0.392 (3.7E-09) | 0.483 (5.7E-13) | 0.481 (6.5E-13) | 0.480 (7.7E-13) |
| 9 | 0.432 (1.0E-10) | 0.293 (7.7E-06) | 0.475 (1.2E-12) | 0.496 (1.3E-13) | 0.494 (1.6E-13) |
| 10 | 0.380 (1.0E-08) | 0.306 (3.4E-06) | 0.401 (1.7E-09) | 0.447 (2.3E-11) | 0.448 (2.1E-11) |
| 11 | 0.376 (1.5E-08) | 0.292 (8.3E-06) | 0.411 (6.9E-10) | 0.448 (2.1E-11) | 0.445 (2.7E-11) |
| 12 | 0.330 (5.7E-07) | 0.127 (3.1E-02) | 0.416 (4.4E-10) | 0.446 (2.5E-11) | 0.447 (2.2E-11) |

## C.4 Perturbation: pretraining seeds and finetuning seeds of BERT medium

Rank correlation scores are low across the board for this task, suggesting that it is difficult for all existing dissimilarity measures, regardless of the layer within a network. Results on the Antonymy stress test for Spearman $\rho$ and Kendall's $\tau$ are in Tables 11 and 12, respectively. Results on the Numerical stress test for Spearman $\rho$ and Kendall's $\tau$ are in Tables 13 and 14, respectively.

Table 11: Layer-wise Spearman $\rho$ results for perturbing pretraining seed and finetuning seed, and assessing functionality through the Antonymy stress test

| Layer | Procrustes | CKA | PWCCA | $\bar{\rho}_{\mathrm{CCA}}$ | $R^2_{\mathrm{CCA}}$ |
|---|---|---|---|---|---|
| 1 | 0.252 (5.7E-03) | 0.241 (7.8E-03) | 0.168 (4.7E-02) | 0.305 (1.0E+00) | 0.327 (1.0E+00) |
| 2 | 0.213 (1.7E-02) | 0.145 (7.5E-02) | 0.131 (9.7E-02) | 0.047 (6.8E-01) | 0.031 (6.2E-01) |
| 3 | 0.260 (4.5E-03) | 0.262 (4.2E-03) | 0.208 (1.9E-02) | 0.137 (9.1E-01) | 0.111 (8.6E-01) |
| 4 | 0.260 (4.5E-03) | 0.265 (3.8E-03) | 0.265 (3.8E-03) | 0.276 (1.0E+00) | 0.254 (9.9E-01) |
| 5 | 0.273 (3.0E-03) | 0.302 (1.1E-03) | 0.278 (2.5E-03) | 0.339 (1.0E+00) | 0.310 (1.0E+00) |
| 6 | 0.330 (3.9E-04) | 0.280 (2.4E-03) | 0.346 (2.1E-04) | 0.313 (1.0E+00) | 0.304 (1.0E+00) |
| 7 | 0.271 (3.2E-03) | 0.315 (7.1E-04) | 0.111 (1.4E-01) | 0.091 (8.2E-01) | 0.090 (8.1E-01) |
| 8 | 0.084 (2.0E-01) | 0.004 (4.8E-01) | 0.123 (1.1E-01) | 0.204 (9.8E-01) | 0.198 (9.8E-01) |

Table 12: Layer-wise Kendall's $\tau$ results for perturbing pretraining seed and finetuning seed, and assessing functionality through the Antonymy stress test

| Layer | Procrustes | CKA | PWCCA | $\bar{\rho}_{\mathrm{CCA}}$ | $R^2_{\mathrm{CCA}}$ |
|---|---|---|---|---|---|
| 1 | 0.199 (1.7E-03) | 0.171 (5.9E-03) | 0.126 (3.3E-02) | 0.244 (1.0E+00) | 0.243 (1.0E+00) |
| 2 | 0.179 (4.3E-03) | 0.123 (3.5E-02) | 0.118 (4.2E-02) | 0.061 (8.1E-01) | 0.042 (7.3E-01) |
| 3 | 0.185 (3.3E-03) | 0.186 (3.2E-03) | 0.139 (2.0E-02) | 0.110 (9.5E-01) | 0.096 (9.2E-01) |
| 4 | 0.187 (3.0E-03) | 0.191 (2.6E-03) | 0.188 (2.9E-03) | 0.206 (1.0E+00) | 0.193 (1.0E+00) |
| 5 | 0.192 (2.4E-03) | 0.194 (2.2E-03) | 0.202 (1.5E-03) | 0.267 (1.0E+00) | 0.242 (1.0E+00) |
| 6 | 0.236 (2.7E-04) | 0.197 (1.9E-03) | 0.252 (1.1E-04) | 0.229 (1.0E+00) | 0.221 (1.0E+00) |
| 7 | 0.189 (2.8E-03) | 0.217 (7.3E-04) | 0.091 (9.1E-02) | 0.081 (8.8E-01) | 0.082 (8.9E-01) |
| 8 | 0.061 (1.9E-01) | -0.000 (5.0E-01) | 0.101 (6.9E-02) | 0.155 (9.9E-01) | 0.150 (9.9E-01) |

Table 13: Layer-wise Spearman $\rho$ results for perturbing pretraining seed and finetuning seed, and assessing functionality through the Numerical stress test

| Layer | Procrustes | CKA | PWCCA | $\bar{\rho}_{\mathrm{CCA}}$ | $R^2_{\mathrm{CCA}}$ |
|---|---|---|---|---|---|
| 1 | 0.137 (8.7E-02) | 0.108 (1.4E-01) | 0.107 (1.4E-01) | 0.072 (7.6E-01) | 0.072 (7.6E-01) |
| 2 | -0.012 (5.5E-01) | 0.060 (2.8E-01) | 0.062 (2.7E-01) | 0.004 (5.1E-01) | 0.001 (5.0E-01) |
| 3 | -0.059 (7.2E-01) | 0.011 (4.6E-01) | -0.031 (6.2E-01) | -0.060 (2.8E-01) | -0.056 (2.9E-01) |
| 4 | 0.041 (3.4E-01) | 0.052 (3.0E-01) | -0.026 (6.0E-01) | -0.101 (1.6E-01) | -0.084 (2.0E-01) |
| 5 | 0.003 (4.9E-01) | 0.131 (9.7E-02) | -0.047 (6.8E-01) | -0.061 (2.7E-01) | -0.061 (2.7E-01) |
| 6 | 0.092 (1.8E-01) | 0.260 (4.5E-03) | -0.029 (6.1E-01) | -0.064 (2.6E-01) | -0.056 (2.9E-01) |
| 7 | 0.164 (5.2E-02) | 0.250 (6.1E-03) | 0.037 (3.6E-01) | 0.040 (6.5E-01) | 0.040 (6.5E-01) |
| 8 | 0.202 (2.2E-02) | 0.105 (1.5E-01) | 0.175 (4.1E-02) | 0.134 (9.1E-01) | 0.143 (9.2E-01) |

Table 14: Layer-wise Kendall's $\tau$ results for perturbing pretraining seed and finetuning seed, and assessing functionality through the Numerical stress test

| Layer | Procrustes | CKA | PWCCA | $\bar{\rho}_{\mathrm{CCA}}$ | $R^2_{\mathrm{CCA}}$ |
|---|---|---|---|---|---|
| 1 | 0.103 (6.5E-02) | 0.083 (1.1E-01) | 0.074 (1.4E-01) | 0.050 (7.7E-01) | 0.048 (7.6E-01) |
| 2 | -0.010 (5.6E-01) | 0.046 (2.5E-01) | 0.046 (2.5E-01) | 0.006 (5.3E-01) | 0.001 (5.0E-01) |
| 3 | -0.041 (7.3E-01) | 0.014 (4.2E-01) | -0.018 (6.0E-01) | -0.047 (2.5E-01) | -0.047 (2.4E-01) |
| 4 | 0.031 (3.2E-01) | 0.038 (2.9E-01) | -0.020 (6.2E-01) | -0.076 (1.3E-01) | -0.065 (1.7E-01) |
| 5 | 0.005 (4.7E-01) | 0.086 (1.0E-01) | -0.031 (6.8E-01) | -0.042 (2.7E-01) | -0.042 (2.7E-01) |
| 6 | 0.060 (1.9E-01) | 0.175 (5.1E-03) | -0.020 (6.2E-01) | -0.050 (2.3E-01) | -0.046 (2.5E-01) |
| 7 | 0.112 (4.9E-02) | 0.168 (6.8E-03) | 0.030 (3.3E-01) | 0.019 (6.1E-01) | 0.024 (6.4E-01) |
| 8 | 0.131 (2.7E-02) | 0.063 (1.8E-01) | 0.125 (3.3E-02) | 0.099 (9.3E-01) | 0.103 (9.4E-01) |

# D  CIFAR-10C subtask-wise results

Table 15: Results for perturbing training seed and assessing functionality through CIFAR-10C

| Table 16: Spearman $\rho$ results | | | Table 17: Kendall $\tau$ results | | |
|---|---|---|---|---|---|
| Corruption | Procrustes | CKA | Corruption | Procrustes | CKA |
| gaussian_noise | 0.083 | 0.076 | gaussian_noise | 0.057 | 0.050 |
| shot_noise | 0.171 | 0.161 | shot_noise | 0.118 | 0.110 |
| impulse_noise | 0.104 | 0.083 | impulse_noise | 0.070 | 0.055 |
| defocus_blur | -0.025 | 0.021 | defocus_blur | -0.016 | 0.013 |
| glass_blur | 0.082 | 0.073 | glass_blur | 0.057 | 0.047 |
| motion_blur | 0.033 | 0.035 | motion_blur | 0.021 | 0.022 |
| zoom_blur | -0.023 | 0.020 | zoom_blur | -0.014 | 0.013 |
| snow | 0.087 | 0.060 | snow | 0.059 | 0.042 |
| frost | -0.062 | -0.081 | frost | -0.046 | -0.059 |
| fog | -0.029 | -0.039 | fog | -0.020 | -0.025 |
| brightness | 0.122 | 0.110 | brightness | 0.084 | 0.077 |
| contrast | -0.225 | -0.145 | contrast | -0.158 | -0.102 |
| elastic_transform | 0.137 | 0.122 | elastic_transform | 0.094 | 0.085 |
| pixelate | 0.118 | 0.098 | pixelate | 0.081 | 0.066 |
| jpeg_compression | 0.149 | 0.102 | jpeg_compression | 0.103 | 0.070 |
| speckle_noise | 0.028 | 0.033 | speckle_noise | 0.019 | 0.022 |
| gaussian_blur | 0.149 | 0.141 | gaussian_blur | 0.102 | 0.095 |
| spatter | 0.089 | 0.079 | spatter | 0.059 | 0.053 |
| saturate | 0.143 | 0.135 | saturate | 0.100 | 0.096 |
| Average | 0.060 | 0.057 | Average | 0.041 | 0.038 |

# E  Bootstrap significance testing for changing fine-tuning seeds

To assess whether the differences between rank correlations are statistically significant in the experiments varying finetuning seed and comparing functional behavior on the OOD HANS dataset, we conduct bootstrap resampling. Concretely, for every pair of metrics and every layer depth, we do the following:

- Sample 100 models with replacement, and collect their representations at the specified layer depth
- Let the reference $A$ be the representation corresponding to the sampled model with maximum accuracy at that depth
- Compute the dissimilarities between $A$ and the 100 sampled representations
- Compute the Kendall's $\tau$ and Spearman's $\rho$ rank correlations for Orthogonal Procrustes, CKA, and PWCCA
- Record $\rho$(Procrustes) - $\rho$(CKA), $\rho$(PWCCA) - $\rho$(CKA), and $\rho$(PWCCA) - $\rho$(Procrustes), and the same pairwise differences for Kendall's $\tau$.
- Repeat the above 2000 times

This gives us bootstrap distributions for the differences in rank correlations, and we may compute the 95% confidence intervals for these distributions. When the confidence interval does not overlap with 0, we conclude that the difference in rank correlation is statistically significant. The figures below show the results for each layer. We see that in the deeper layers of the network (layers 8-12), PWCCA has statistically significantly higher rank correlation than Orthogonal Procrustes, which in turn has statistically significantly higher rank correlation than CKA. In earlier layers, results are sometimes statistically significant, but not always.

Figure 4: Bootstrap comparison of $\rho$ between metrics, layers 1-4

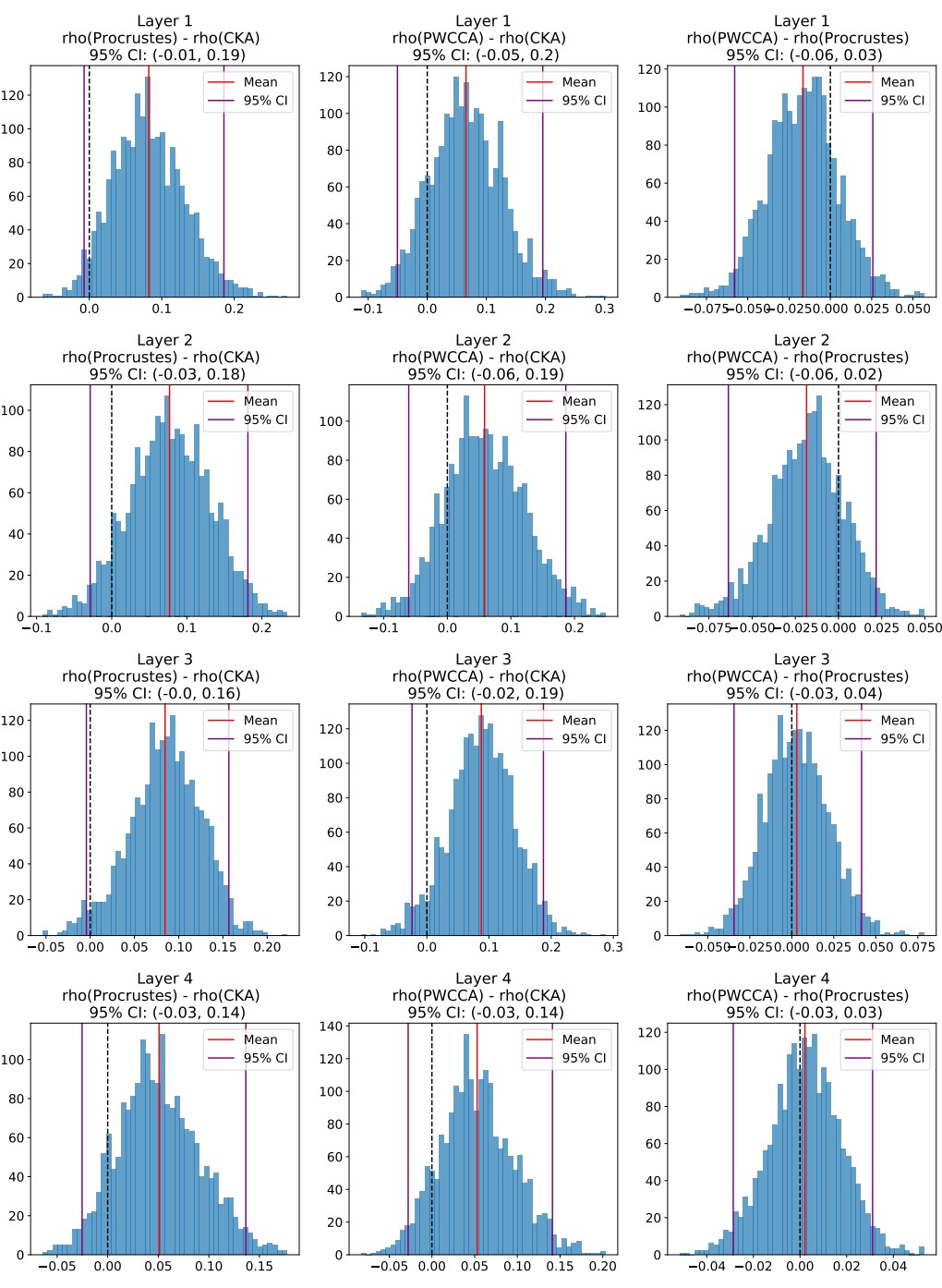

Figure 5: Bootstrap comparison of $\rho$ between metrics, layers 5-8

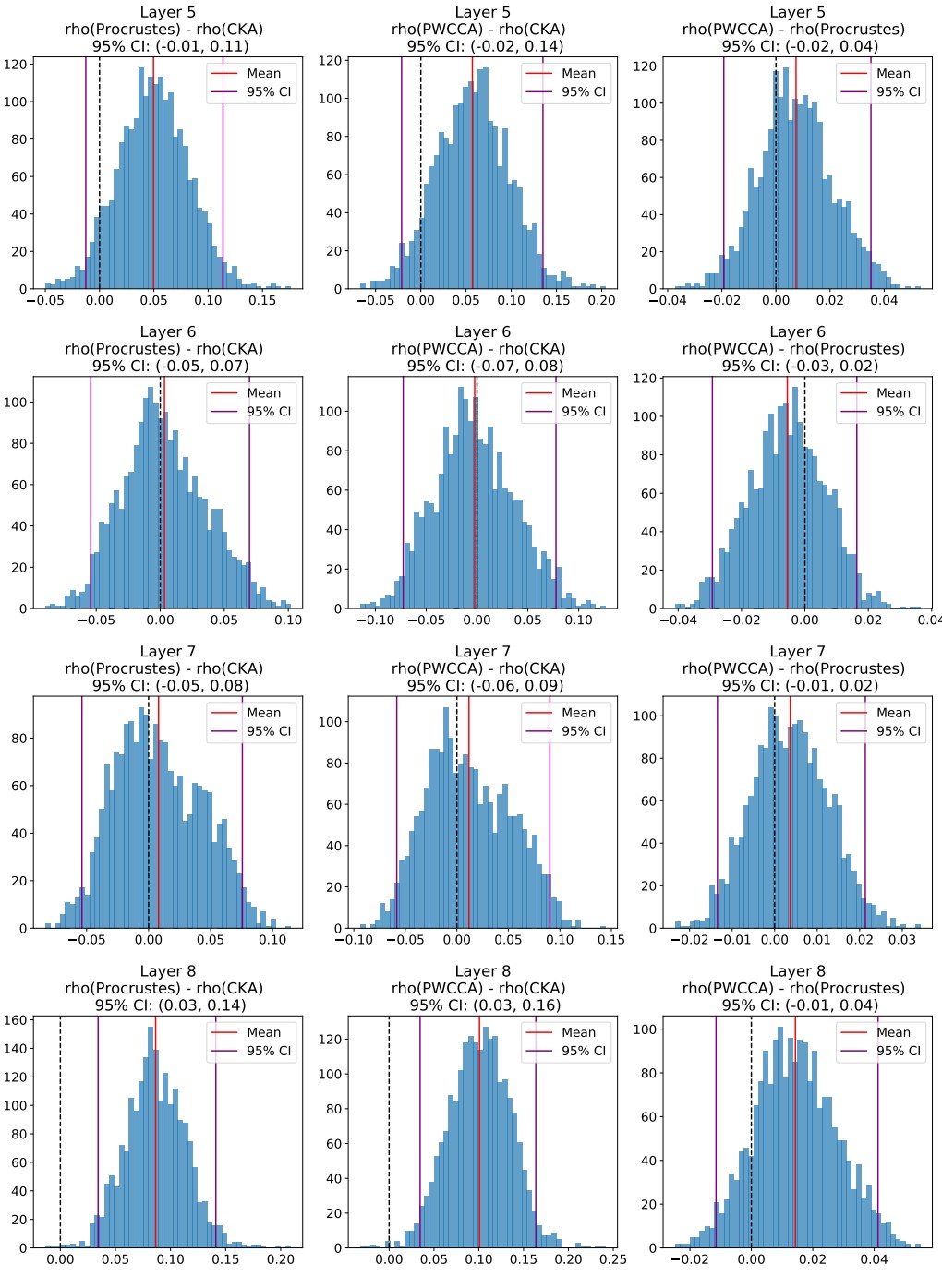

Figure 6: Bootstrap comparison of $\rho$ between metrics, layers 9-12

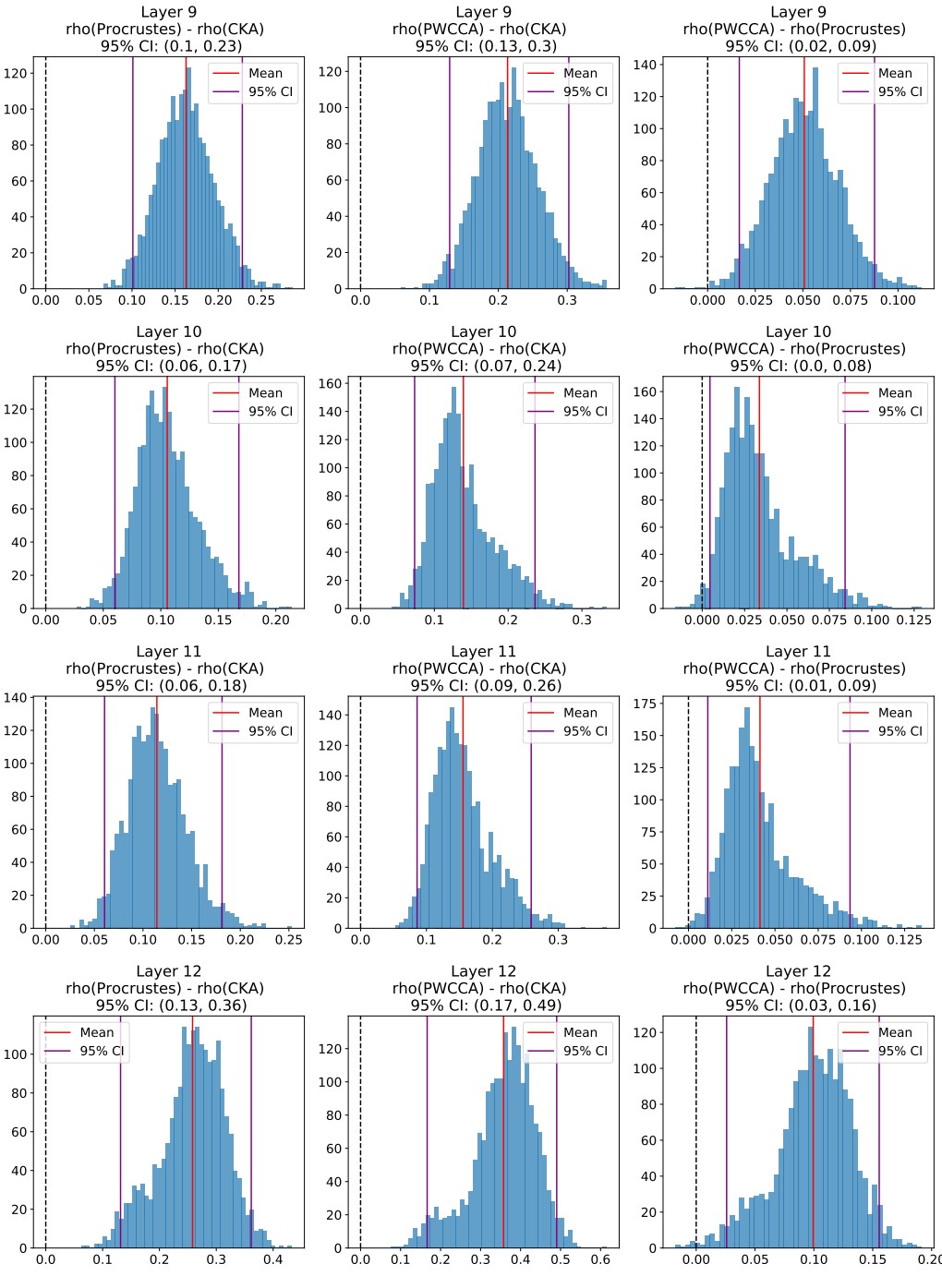

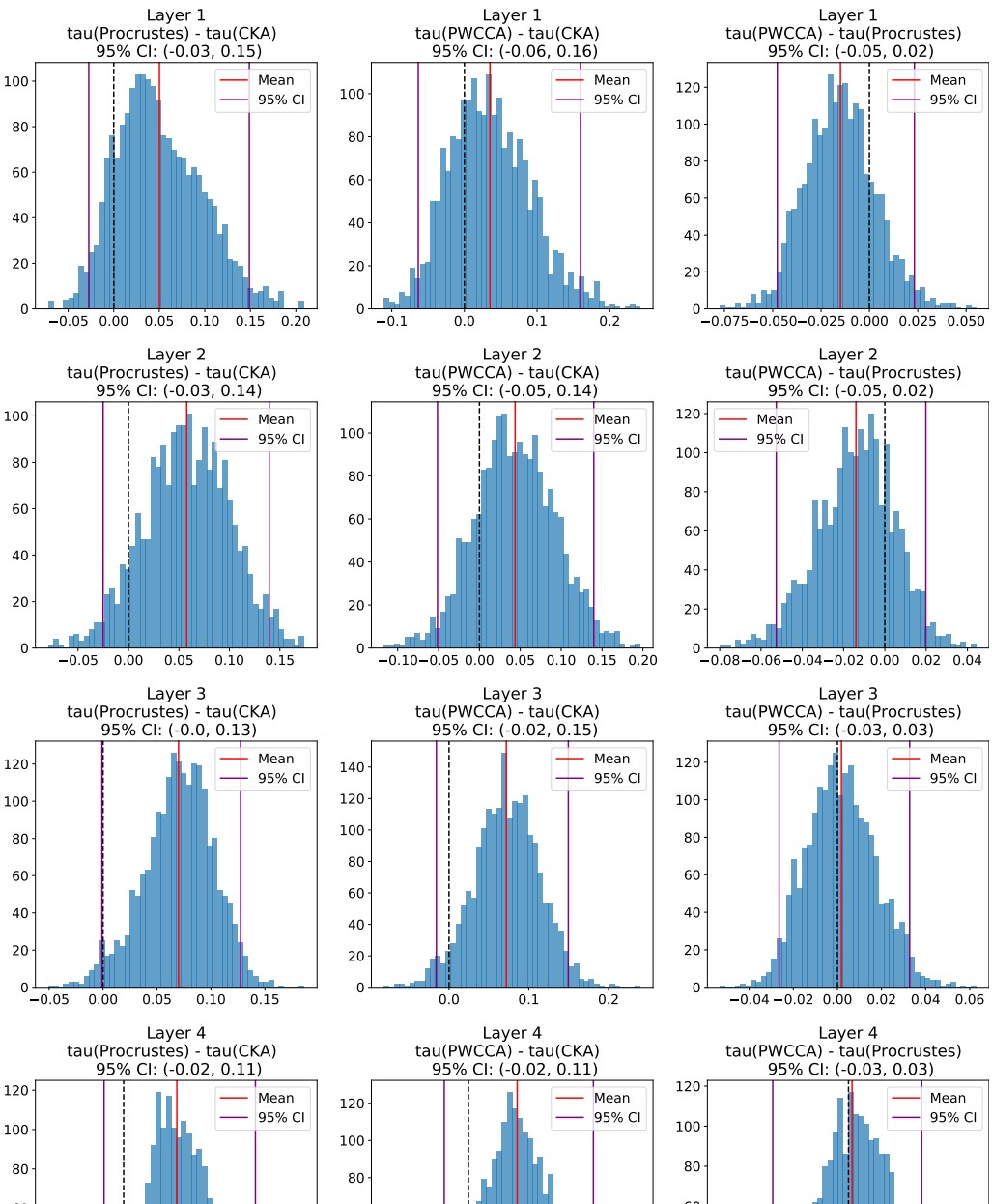

Figure 7: Bootstrap comparison of $\tau$ between metrics, layers 1-4

Figure 8: Bootstrap comparison of $\tau$ between metrics, layers 5-8

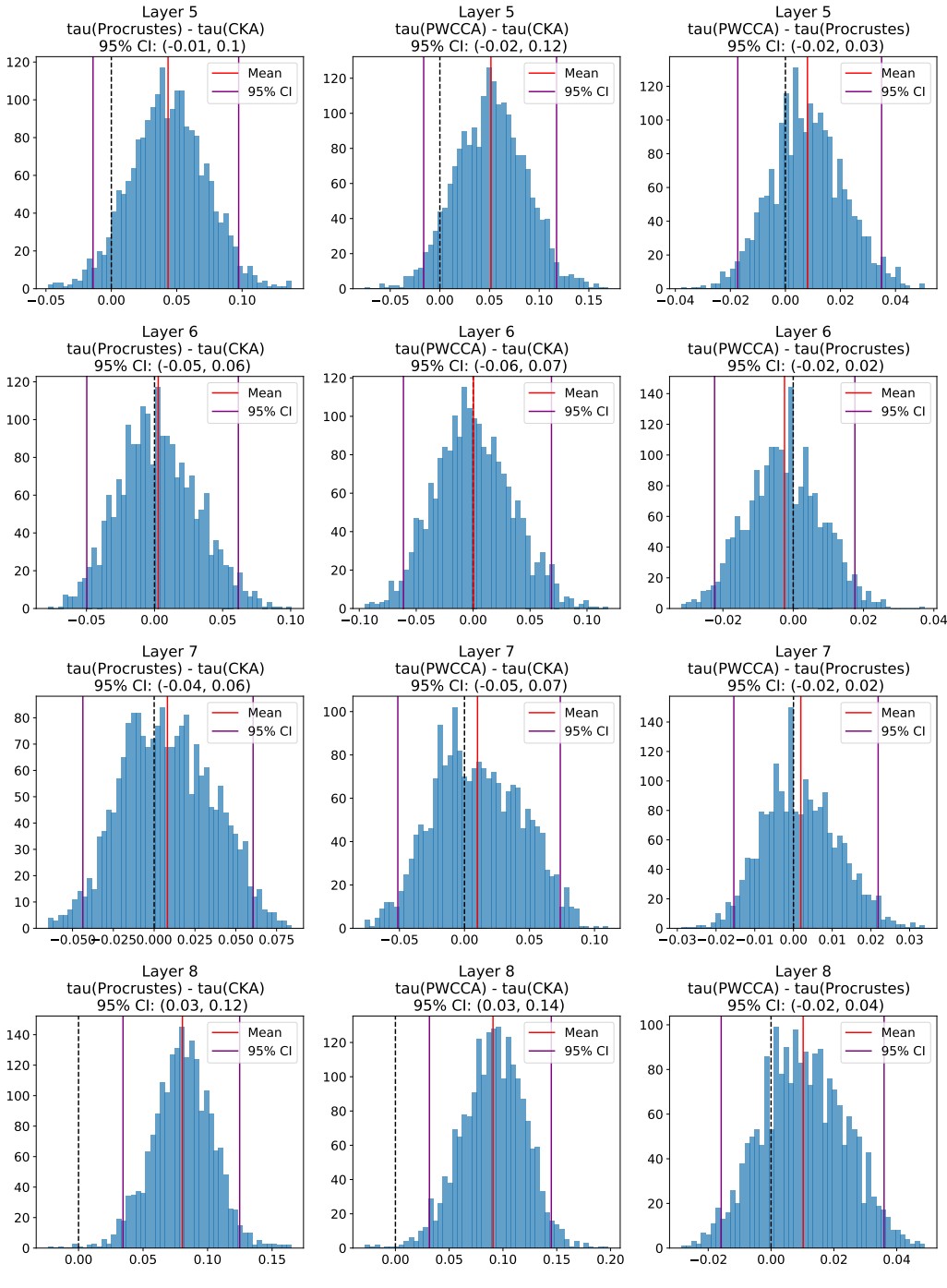

Figure 9: Bootstrap comparison of $\tau$ between metrics, layers 9-12

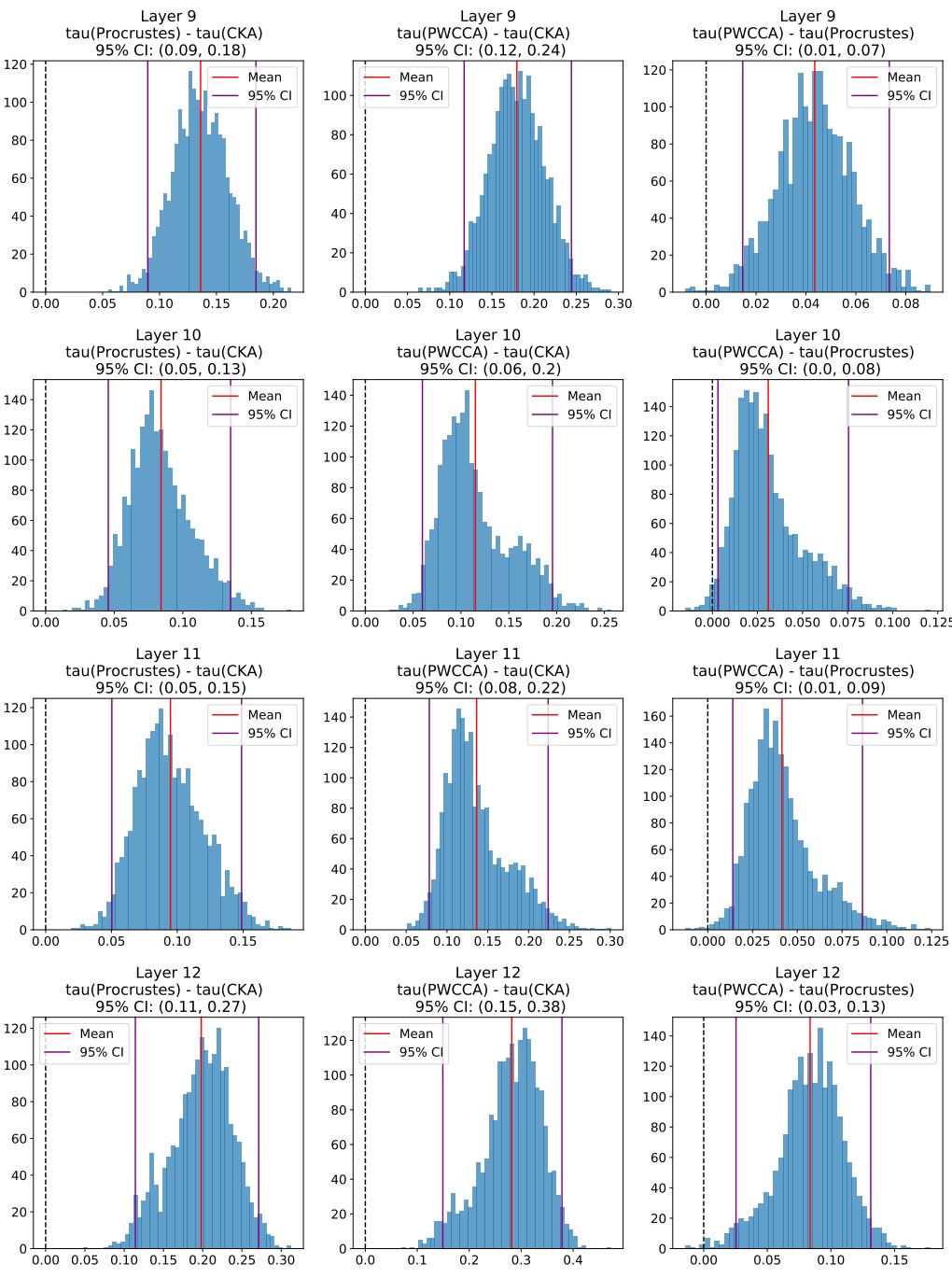