# OpenReview forum: "Grounding Representation Similarity Through Statistical Testing"
_NeurIPS.cc/2021/Conference — NeurIPS 2021 Poster_

### Official Review · Reviewer_swz6 · 2021-07-12

**Rating:** 6
**Confidence:** 2

**Summary:**

The authors present a number of benchmarks and stress tests for similarity measures of learned representations of neural networks. Three similarity metrics are compared for their sensitivity and specificity of the similarity measure under a number of conditions, such as out-of-domain data or low rank approximations of hidden layer activation. Empirical results suggest that there is no clear trend for one or the other similarity metric that would be robust against all types of tests proposed.

**Update**
After reading the authors responses I found that they addressed most of my concerns sufficiently. My main concern was that CCA a) is not evaluated with cross-validation and instead canonical correlations are computed on training data and b) that the CCA projections are not regularized. These two points together can lead to overfitted CCA projections and biased estimates of the canonical correlations, especially in the settings addressed in the present study. This could explain the poor results of CCA in the comparisons. Th

The authors responded that this is how the PWCCA method was applied and evaluated and the overfitted and not-cross-validated canonical correlations are a 'feature' of PWCCA. I wouldn't agree to that personally, but I think it's a valid point -- after all many researchers do not evaluate unsupervised methods on held-out test data, so there's probably some value in examining and comparing how these (wrong and biased) metrics computed on training data behave.

So I think it's a valuable contribution and I increased my score accordingly.



**Ethics Review Area:**

["I don’t know"]

**Limitations And Societal Impact:**

yes

**Main Review:**



**Originality**:

The probing tasks proposed are novel to the best of my knowledge, however they are a combination of previously proposed methods. It’s an important contribution to evaluate these different methods, and the idea to compare these methods under a variety of different conditions is also interesting.

**Quality**:

The submission is technically sound for most parts, but there are some technical details and  some motivations/intuitions that could be made clearer.

For instance in line 99 the authors write that

*We find that PWCCA performs far better than …*, here it would be helpful to see what performance is measured.

If canonical correlations are not “working right” it could also be due to how they are evaluated. With high dimensional low rank data sets (and some large layers might be of that kind), CCA requires regularisation, and it makes sense to avoid overfitted canonical correlations (ccs) by evaluating the ccs on held-out data, just like with any supervised model. In line 302 the authors also suggest that CCA might have overfitted, but if it’s really overfitted it would not result in high held-out ccs. So both the hyper parameter optimisation for the CCA regularisation parameter as well as the evaluation on held out data might change the CCA results and I would assume the ccs are more trustworthy when cross-validated.
That said, the other alternative similarity measures are probably more attractive as they do not require fitting parameters.

The idea with most probing tests makes a lot of sense, but I’m not sure I understand what exactly is measured in the experiments with removal of PC components, or rather how that measure would help to understand something about representations - it's hard to relate that transformation to something that happens to networks in applications. A minor general remark on the selection of components, I think it could be helpful to discard components based on the amount of variance they explain, not their absolute number.

In general it seems the authors often use the term “performance” and it is not really clear what is meant by that. It seems the methods are better or worse on different tasks, but it could be better explained what exactly is measured across all tasks, and what we can conclude from the results, beyond the observations for the single experiments.

Also there are number of packages for perturbations and augmentations that would be worth integrating into these tests.

**Clarity**:

The manuscript is well written and structured, but there could be a bit more structure in the interpretation of results or the experimental design. I was asking myself what the results imply and what specificity and sensitivity results tell us about the networks if the probing tasks are so specific and lead to different results for most similarity metrics. Maybe one could evaluate just robustness of the similarity metrics under perturbations. Or maybe there is a way to add more probing tests or modify the metric to see more consistent differences between similarity metrics. In the end one is interested in things like which metric should I use to evaluate which aspect.


**Significance**:

Developing tests to evaluate is a relevant research direction and the tests proposed in this work are an important contribution. There are some aspects about the evaluation that could be improved and the heterogeneity of the empirical results suggest that it could be difficult to draw general conclusions about neural representations.

**Time Spent Reviewing:**

3

---

> ### Author Response · Authors · 2021-08-11
> **Response to Reviewer swz6**
>
> We thank the reviewer for their comments and feedback. We appreciate that they found the work to be an important and interesting contribution. Below we address some of the questions raised in the review:
>
> >in line 99 the authors write that ‘We find that PWCCA performs far better than …’, here it would be helpful to see what performance is measured.
>
> In our revised version we will clarify that the performance measure is the rank correlation score described in Section 4, and that full results can be found in the Appendix that show PWCCA achieving higher rank correlations than the other CCA-based methods.
>
> >If canonical correlations are not “working right” it could also be due to how they are evaluated. With high dimensional low rank data sets (and some large layers might be of that kind), CCA requires regularisation, and it makes sense to avoid overfitted canonical correlations (ccs) by evaluating the ccs on held-out data, just like with any supervised model.
>
> In previous literature using CCA methods for representation similarity, held-out data has not been used for regularization; instead, techniques such as projection weighting have been proposed (thus our choice to study PWCCA). We intended to fairly compare methods in the literature with our benchmark, and matched the implementations in these prior works. However, the suggestion to use held-out data is a very interesting one, and perhaps a new dissimilarity measure could be designed with that in mind, that will do even better on our benchmark tasks -- this would be an interesting direction for future work.
>
> >The idea with most probing tests makes a lot of sense, but I’m not sure I understand what exactly is measured in the experiments with removal of PC components, or rather how that measure would help to understand something about representations - it's hard to relate that transformation to something that happens to networks in applications. A minor general remark on the selection of components, I think it could be helpful to discard components based on the amount of variance they explain, not their absolute number.
>
> We used removal of principal components as a way to simulate loss in information. Since representations are matrices of activations, removing principal components to create rank-$k$ approximations of those matrices, for a series of values of $k$, seemed like a natural way to smoothly remove information. There is also recent network distillation work that computes low-rank approximations of network representations to train a smaller “student” network to match the performance of a larger “teacher” network [1].
>
> >In general it seems the authors often use the term “performance” and it is not really clear what is meant by that. It seems the methods are better or worse on different tasks, but it could be better explained what exactly is measured across all tasks, and what we can conclude from the results, beyond the observations for the single experiments.
>
> Each task is intended to highlight a different functionality that a good dissimilarity measure should be responsive to. To obtain a single summary of performance, we averaged together the rank correlation for the different tasks in Table 1. The results are below:
>
> | Procrustes   $\rho$   | Procrustes   $\tau$   | CKA $\rho$ | CKA $\tau$ | PWCCA $\rho$| PWCCA $\tau$|
> | ----------- | ----------- | ------------- |------------- |------------- |------------- |
> |**0.580** |**0.447** |0.557 |0.426 |0.544 |0.413|
>
> We find that Orthogonal Procrustes has the highest average score by a significant margin (larger than the difference between CKA and PWCCA, the next two best methods). While it is true that sometimes other methods perform best on specific subtasks, and thus the average might mask important variation, it is also the case that Procrustes is close to the best on every individual subtask as well (i.e., second best at worst, and with a score closer to the best than the third best).
>
>
> References:
>
> [1] Lee, S. H., Kim, D. H., & Song, B. C. (2018). Self-supervised knowledge distillation using singular value decomposition. In Proceedings of the European Conference on Computer Vision (ECCV) (pp. 335-350).

---

> > ### Comment · Reviewer_swz6 · 2021-08-11
> > **On regularization and cross-validation of canonical correlations**
> >
> > > In previous literature using CCA methods for representation similarity, held-out data has not been used for regularization
> >
> > I think it's important to consider regularization and cross-validation separately here. Canonical Correlations should be evaluated on held-out data, otherwise the correlations can be arbitrarily high depending on the task/data/dimensionality. The fact that other studies have not cross-validated the canonical correlations is unfortunate but not an argument to not do this right. As for the regularization, this is just a technique to prevent the overfitting of canonical correlations. If this is not done, overfitting is more likely to happen - but if the canonical correlations are evaluated on the training data, this won't be detected.

---

> > > ### Author Response · Authors · 2021-08-12
> > > **Re: regularization and cross-validation of canonical correlations**
> > >
> > > Thank you for your engagement and your thoughtful reply!
> > >
> > > We aren't sure if you are claiming that lack of cross-validation invalidates the paper’s results, or that it makes PWCCA a bad method. If it is the former, we think you might be misunderstanding the setup of the paper, as the use or non-use of cross-validation doesn't affect the validity of the results--all similarity measurements are made on a test set already, per standard practice. If the latter, we agree, and in fact PWCCA did perform the worst under the averaged scores. But our point was to evaluate similarity metrics in the literature--your proposed metric might perform better, but it would be creating a new metric, and in particular it would be a different metric than PWCCA. If you would find it helpful for us to implement and report results for your proposal, we are happy to do so (but we have some clarification questions below on what you have in mind).
> > >
> > > All the canonical correlation coefficients in the paper are computed on networks’ representations of test set data, not training data, so for cross-validation did you mean that we should cross-validate on subsets of the test data, as follows?:
> > > 1. Split the test data in half (call them subsets S and T).
> > > 2. Compute canonical correlation coefficients $\rho_S$ and orthogonal bases ($\omega_S$, $\omega_S’$) from applying CCA to two networks’ representations of S.
> > > 3. Using the orthogonal bases ($\omega_S$, $\omega_S’$), linearly transform the two networks’ representations of T, and compute the correlations $\rho_T$ between the transformed representations of T.
> > > 4. Report $\rho_T$ as the cross-validated canonical correlation coefficients.
> > > 5. Average the correlation coefficients together (perhaps with weighting), and report the scalar result as the similarity score.
> > >
> > > Also, for regularization, did you have in mind the ridge regularization as in [1], possibly with additional cross-validation to select the regularization parameter as in [2], or another form of regularization?
> > >
> > > References:
> > >
> > > [1] Vinod HD: Canonical ridge and econometrics of joint production. Journal of Econometrics 1976, 4(2):147–166. 10.1016/0304-4076(76)90010-5
> > >
> > > [2] Golugula, Abhishek, et al. "Supervised regularized canonical correlation analysis: integrating histologic and proteomic measurements for predicting biochemical recurrence following prostate surgery." BMC bioinformatics 12.1 (2011): 1-13.

---

> > > > ### Comment · Reviewer_swz6 · 2021-08-12
> > > > **Cross-validation of canonical correlations**
> > > >
> > > > First of all thanks for the detailed response - I'm fully aware that there are many things that are being done just because the community has been doing that often.
> > > >
> > > > I guess the reason why I mentioned the overfitting aspects related to CCA is that I've been using CCA a lot and I've been evaluating CCA like in your submission - until a reviewer pointed me to the fact that CCA, like all supervised methods (and also unsupervised methods), should be evaluated on held out data. Switching to cross-validation and tuning the regularization parameters of CCA helped a lot to get more authentic and unbiased estimated of the canonical correlations.
> > > >
> > > > > We aren't sure if you are claiming that lack of cross-validation invalidates the paper’s results, or that it makes PWCCA a bad method.
> > > >
> > > > I wouldn't say that CCA is a bad method, if it's not evaluated right. It's just a biased metric if one is interested in generalization performance. Maybe it's helpful to think of (PW)CCA as a standard supervised method: I guess one wouldn't argue that it's a bad method because it's evaluated on the training data.
> > > >
> > > > What I'd suggest is to train the CCA projections on training data and evaluate the projections (i.e. compute the canonical correlations) on test data. I'm terribly sorry if this was actually done and I just missed it.
> > > >
> > > > And as for the regularization: yes, I meant Tikhonov regularization, i.e. just adding a ridge to the (auto)-covariance matrices in the denominator of the Rayleigh quotient (the right hand side of the generalized eigenvalue equation) - many CCA implementations or the eigensolvers used therein do that anyways by default with a small ridge of height 1e-5 just for numerical stability. But there are other regularizers as well.
> > > >
> > > > Both of these measures could help to unbias the canonical correlation estimates and potentially change the rankings and assessment of the methods.

---

> > > > > ### Author Response · Authors · 2021-08-14
> > > > > **Cross-validation details**
> > > > >
> > > > > Thank you for the detailed reply!
> > > > >
> > > > > Based on this discussion, we think you are proposing the following method as a cross-validated, regularized version of PWCCA (CV-PWCCA for short). If this description matches what you have in mind, we would be happy to implement it and report the results; if we’ve misunderstood, please let us know what you would like to see.
> > > > >
> > > > > ---
> > > > > Algorithm: CV-PWCCA
> > > > >
> > > > > **Inputs**: Two $p \times n$ matrices of activations, $A$ and $B$
> > > > >
> > > > > **Hyperparameters**: Fixed scalar $\lambda$ that determines the strength of Tikhonov regularization
> > > > >
> > > > > **Step 1: Split representations into train and test.** Split $A$ into $A_{train}$ and $A_{test}$, each $p \times  n/2$ matrices, and $B$ into $B_{train}$ and $B_{test}$ similarly.
> > > > >
> > > > > **Step 2: Compute regularized CCA projections on train data.** Add $\lambda * I$ to the autocovariance matrices in the generalized eigenvalue equation for regularization, and solve to find the projection vectors $\omega_{A_{train}}$, $\omega_{B_{train}}$.
> > > > >
> > > > > **Step 3: Compute canonical correlations on test data.** For $1 \leq i \leq p$, compute $\rho_i = \frac{\left< {\omega_{A_{train}}^i}^\top A_{test}, {\omega_{B_{train}}^i}^\top B_{test} \right>} {||{\omega_{A_{train}}^i}^\top A_{test} ||    \cdot   ||  {\omega_{B_{train}}^i}^\top B_{test}  ||} $, so we have $p$ cross-validated canonical correlation coefficients, $\rho_i$.
> > > > >
> > > > > **Step 4: Compute the projection weighted mean of the correlation coefficients.** Compute weights $\alpha_i = \sum_j | \left< h_i, a_j \right>|$ where $a_j$ is the $j$th row of $A_{test}$ and $h_i =  {\omega_{A_{train}}^i}^\top A_{test}$ is the projection of $A_{test}$ onto the $i$th canonical direction. Use $\alpha_i$ to compute the weighted mean of $\rho_i$.
> > > > >
> > > > > **Step 5: Return the scalar result as CV-PWCCA(A, B).**
> > > > >
> > > > > ---
> > > > > Note that the representations A and B are embeddings of “test data” already, as they have not been seen by the neural networks before. We further split them into train and test subsets above, for the purposes of CCA cross-validation. The other measures, CKA and Orthogonal Procrustes, do not need cross-validation and do not need to do this splitting.

---

### Official Review · Reviewer_SGnL · 2021-07-14

**Rating:** 7
**Confidence:** 4

**Summary:**

The work is motivated by a desire to better understand representational similarity metrics and ensure they obey certain desirable properties. Specifically, representational similarity measures should be sensitive changes that affect functional behavior, and specific (invariant) to changes that do not affect functional behavior. The authors operationalize these desiderata as 1) sensitivity to deletions of principal components in activation matrices, and 2) invariance to variations between models that arise from different random seeds. They find that CCA-based methods are less specific than CKA and Orthogonal Procrustes Analysis (OPA), while CKA is less sensitive than CCA and OPA. The authors then establish “benchmarks” by examining how certain variables and interventions (e.g. deleting principal components, changes in random seed, fine-tuning) affect the correlation between representational similarity metrics and performance metrics (mainly linguistic probes). The authors find that the benchmarks generally recapitulate the initial findings re: specificity and sensitivity. OPA seems to fall in between CKA and CCA with respect to the trade-off between sensitivity and specificity, leading the authors to encourage its adoption.

**Limitations And Societal Impact:**

The authors state “A limitation of this work is that we only consider a handful of functional behaviors”, which I think is true, and I provide a number of suggestions in the “Quality” section of the review.

**Main Review:**

**Excellent response. Score updated from 3 to 7. Details below**

**Originality: Are the tasks or methods new? Is the work a novel combination of well-known techniques? (This can be valuable!) Is it clear how this work differs from previous contributions? Is related work adequately cited?**

It takes a set of established representational similarity methods, prescribes desirable operating principles, and attempts to understand and benchmark them with regards to the principles. The clear articulation of desirable operating principles are somewhat novel, and the principal component deletion and benchmarking components of this work are novel combinations of known techniques. The effects of random seeds/initialization on representational similarity metrics has already been examined by Kornblith et al. (2019), though they did not examine OPA. Otherwise, related work appears adequately cited.

**Quality: Is the submission technically sound? Are claims well supported (e.g., by theoretical analysis or experimental results)? Are the methods used appropriate? Is this a complete piece of work or work in progress? Are the authors careful and honest about evaluating both the strengths and weaknesses of their work?**

In its current form, I think the scope of the paper is limited. The analyses are limited to transformer models and linguistic probing tasks. Extending the analyses to CNNs and even vision transformers (and multiple datasets) would allow the authors to make stronger claims about generality and increase the relevance of this work. This could be accomplished with layer-wise probing tasks similarly to with the language models. Comparisons should also be made between different model architectures.

More generally, I am curious about the following issue: The authors state that “Metrics should have specificity against random initialization” (L41-42). If two different initializations drawn from the same distribution result in two networks with very different behavior (i.e. different outputs for the same inputs), do we want a representational similarity metric to detect this? My assumption is yes, but perhaps I’m mistaken. This should be discussed.

The authors state “Neural network representations trained on the same data but from different random initializations are similar” (L125-126). Is this axiomatic, or a claim that representational similarity measures are intended to test? If it’s the latter, can you expand on what Kornblith et al. (2019) report?

I believe it’s implicit that “similarity” between networks always means representational similarity, but I think it’s important that you explicitly dissociate representational vs. functional similarity. If two networks have very different representations but very similar outputs, would these networks be “similar”? By the definition of Sundararajan et al. (2018; Axiomatic Attribution for Deep Networks), these networks are functionally equivalent. To this end, comparing distributions (of outputs/classifier layers) could be more informative than just comparing task accuracies.

Doing more probe-representation benchmarks seems like an easy way to strengthen your work, as probing tasks are cheap to run and there are a lot of them.

**Clarity: Is the submission clearly written? Is it well organized? (If not, please make constructive suggestions for improving its clarity.) Does it adequately inform the reader? (Note that a superbly written paper provides enough information for an expert reader to reproduce its results.)**

The submission is sufficiently clear and organized, but I have a few questions and suggestions:

Looking at Table 1, CKA seems seems to be most consistent, yet the authors state “the classical Orthogonal Procrustes transform attained consistently good performance” (ln 289-290). Can these opinions be reconciled? Perhaps by quantification?

The results in Lines 169-172 should be presented in a figure, e.g. plotting PC-deletion vs. accuracy curves.

More information should be provided about the rows in Table 1. For example, the OOD conditions should be explicitly labeled as such.

**Significance: Are the results important? Are others (researchers or practitioners) likely to use the ideas or build on them? Does the submission address a difficult task in a better way than previous work? Does it advance the state of the art in a demonstrable way? Does it provide unique data, unique conclusions about existing data, or a unique theoretical or experimental approach?
I think the work is well-motivated.**

The authors state “As a community, we need well-chosen formal criteria for evaluating metrics to avoid over-reliance on intuition and the pitfalls of too many researcher degrees of freedom” (L30-31), and I strongly agree with this. But as stated in the “Quality” section of this review, in its current form the work’s primary (and severe) limitation is that it only uses transformer models and language tasks. However, I am optimistic about the potential relevance and impact of this work if my concerns are addressed.


**Time Spent Reviewing:**

4

---

> ### Author Response · Authors · 2021-08-10
> **Response to Reviewer SGnL**
>
> Thank you for your comments and feedback; we appreciate that you are optimistic about the relevance and impact of the work.
>
> Your primary concern was about the diversity of settings studied in the paper, and the lack of experiments on vision models. First, despite being restricted to language models, we believe the current benchmarks are quite diverse: the comparisons involve 3740 distinct representation pairs, extracted from over 200 distinct models, and measure functionality on 5 distinct probe or test sets. More importantly, we did extend the analysis to vision as you suggested, and found qualitatively similar trends (see below for details). If these help address your feeling that the paper is a “clear rejection”, we would be grateful if you considered revising your score.
>
> We ran the following two experiments with vision models:
> 1. A version of our OOD experiments using 50 ResNet-14 models trained on CIFAR-10 with different random initialization. The results are similar to language: as before, Orthogonal Procrustes achieves higher rank correlations than CKA (CCA metrics perform trivially since $n>p$). More details can be found in the comment below titled “Experiment 1: OOD benchmarks in vision models”.
>
> 2. A version of our probing accuracy vs. layer depth experiments, using 10 ResNet-14 models trained on CIFAR-10 with different random initialization. The results are also similar to language: CKA and Procrustes achieve comparable rank correlation scores. More details can be found in the comment below titled “Experiment 2: Probing benchmarks in vision models”.
>
> We will include these results in the final draft, and also plan to include results for other CNN architectures.
>
> Below we address the other concerns you raise:
>
> >The authors state that “Metrics should have specificity against random initialization” (L41-42). If two different initializations drawn from the same distribution result in two networks with very different behavior (i.e. different outputs for the same inputs), do we want a representational similarity metric to detect this? My assumption is yes, but perhaps I’m mistaken. This should be discussed.
>
> We indeed want a representational similarity metric to detect such differences, and that precisely motivates our benchmarks in Section 4. In L205-206 we write that one benefit of our rank correlation procedure, over the Section 3 intuitive tests, is that “some variation in random seed actually does affect accuracy, and the procedure rewards metrics that pick up on this” . We will make this point more clear in the final draft -- in stating the intuition that “metrics should have specificity against random initialization”, we meant to jog intuition about plausibly similar representations, and reference previous sanity checks, such as in Kornblith et al., that expect networks with different initializations to have similar representations at each depth. Then in Section 4 we show that there are still differences across random initialization that a good dissimilarity measure should detect.
>
> >The authors state “Neural network representations trained on the same data but from different random initializations are similar” (L125-126). Is this axiomatic, or a claim that representational similarity measures are intended to test? If it’s the latter, can you expand on what Kornblith et al. (2019) report?
>
> The sentence expresses an intuition about networks that we temporarily took as given, for the purposes of designing the intuitive tests in Section 3. However, as noted above, by Section 4 we note that this is not always true -- networks trained from different initializations can be substantially different, and thus a better way to evaluate dissimilarity measure is through rank correlation scores on our benchmarks.
>
> >I believe it’s implicit that “similarity” between networks always means representational similarity, but I think it’s important that you explicitly dissociate representational vs. functional similarity. If two networks have very different representations but very similar outputs, would these networks be “similar”? By the definition of Sundararajan et al. (2018; Axiomatic Attribution for Deep Networks), these networks are functionally equivalent. To this end, comparing distributions (of outputs/classifier layers) could be more informative than just comparing task accuracies.
>
> We agree that representation similarity and functional similarity are different, and what we are ultimately after is some form of representation similarity. However, representation similarity is itself ill-defined: which directions should matter when determining similarity? At least some dimensions of representations are probably noise (as evidenced by the possibility of distilling networks) and presumably shouldn’t count. Our point therefore is that we need to evaluate proposed metrics for representation similarity using functional similarity--but rather than using a single function, use many functions and datasets as we do here.
>
> >Looking at Table 1, CKA seems to be most consistent, yet the authors state “the classical Orthogonal Procrustes transform attained consistently good performance” (ln 289-290). Can these opinions be reconciled? Perhaps by quantification?
>
> This is an excellent point. To do this, we take the average performance across the different benchmarks in Table 1. The results are:
>
> |Procrustes $\rho$|Procrustes $\tau$|CKA $\rho$|CKA $\tau$|PWCCA $\rho$| PWCCA $\tau$|
> | ---------------------- | ----------- | ------------- |------------- |------------- |------------- |
> |**0.580** |**0.447** |0.557 |0.426 |0.544 |0.413|
>
> This further supports the idea that Orthogonal Procrustes, despite being overlooked in the past, actually performs quite strongly.
>
> >The results in Lines 169-172 should be presented in a figure, e.g. plotting PC-deletion vs. accuracy curves.
>
> We plotted the relevant PC-deletion vs. accuracy curve in Figure 3b (and referenced this in lines 167-168). Since the figure appears on a later page, we can include this PC-deletion vs. accuracy curve earlier in the final draft for easier reference when reading this passage.
>
> >More information should be provided about the rows in Table 1. For example, the OOD conditions should be explicitly labeled as such.
>
> In the revised version we will add more details to the table as suggested.

---

> > ### Author Response · Authors · 2021-08-10
> > **Experiment 1: OOD benchmarks in vision models**
> >
> > We train 50 ResNet-14 models on CIFAR-10 with different random initialization, and assess whether dissimilarity measures correlate with differences in OOD accuracy on each of the 19 datasets in CIFAR-10-C [1], where distortions such as Gaussian noise are added to CIFAR-10 images. To check that the in-distribution training is successful, we note that all models achieve accuracy of $0.91 \pm .01$ on the CIFAR-10 test set. To check that our benchmark contains models with varying functionality, we note that accuracy on the OOD datasets varies significantly -- usually the best and worst performing models have an accuracy difference of 10-20 percentage points.
> >
> > We compute representation dissimilarity on the CIFAR-10 test set (i.e., similar to the HANS experiments in the paper, the dissimilarity measures are using representations of IID data to predict OOD accuracy differences). We compare CKA and Orthogonal Procrustes’ rank correlations (CCA metrics perform trivially since $n>p$). Averaging the results over 14 layers, we see that Procrustes has a higher rank correlation than CKA for most tasks:
> >
> > |OOD task |Procrustes $\rho$ | Procrustes  $\tau$ | 	 CKA $\rho$ | 	CKA $\tau$ |
> > |-------------| -------------| -------------|-------------|-------------|
> > | gaussian_noise       |**0.265** |	**0.202** |	0.221 |	0.164 |
> > | shot_noise |              **0.308** |	**0.226** |	0.234 |	0.171 |
> > | impulse_noise |	     0.227 |	0.153 |	**0.249** |**0.172** |
> > | defocus_blur |    .      **0.203** |**0.141**        |       0.162 |0.105 |
> > | glass_blur |	            0.173 |	0.118 |	**0.213** |	**0.149** |
> > | motion_blur |	            **0.335** |	**0.232** |	0.261 |	0.180 |
> > | zoom_blur |	           **0.332** |	**0.241** |	0.245 |	0.171 |
> > | snow |	            **0.306** |	**0.221** |	0.239 |	0.170 |
> > | frost |	                        **0.325** |	**0.234** |	0.242 |	0.176 |
> > | fog |	                        **0.166** |	**0.120** |	0.133 |	0.094 |
> > | brightness |	            **0.337** |	**0.247** |	0.231 |	0.160 |
> > | contrast |	            **0.273** |	**0.200** |	0.184 |	0.130 |
> > | elastic_transform |	**0.233** |	**0.165** |	0.229 |	0.161 |
> > | pixelate |	            0.254 |	0.175 |	**0.278** |	**0.196** |
> > | jpeg_compression |	**0.281** |	**0.206** |	0.277 |	0.201 |
> > | speckle_noise |	**0.317** |	**0.231** |	0.224 |	0.166 |
> > | gaussian_blur |	**0.248** |	**0.172** |	0.187 |	0.123 |
> > | spatter |	0.243 |	0.174 |	**0.250** |	**0.182** |
> > | saturate |	**0.276** |	**0.202** |	0.187 |	0.135 |
> > |**Average**|	**0.269** |	**0.193**|	0.223|	0.158|
> >
> >
> > The averaging hides the fact that certain layers had high rank correlations, and others had low correlation. The comparison between CKA and Procrustes is consistent across layers, and we can report the results for all 14 layers, if they are of interest. These results reinforce the conclusion that Orthogonal Procrustes performs strongly, capturing functionality in relevant OOD tasks.
> >
> > References:
> >
> > [1] Hendrycks, D., & Dietterich, T. (2019). Benchmarking neural network robustness to common corruptions and perturbations. arXiv preprint arXiv:1903.12261.

---

> > > ### Author Response · Authors · 2021-08-11
> > > **Experiment 2: Probing benchmarks in vision models**
> > >
> > > We train 10 ResNet-14 models on CIFAR-10 with different random initialization, and assess whether dissimilarity measures correlate with differences in probing accuracy across layers and across networks on two probing tasks: CIFAR-100 and Street View House Numbers (SVHN). To check that our benchmark contains models with varying functionality, we note that accuracy on the probing datasets varies significantly -- usually the best and worst performing models have an accuracy difference of 30+ percentage points.
> > >
> > > We compute representation dissimilarity on the CIFAR-10 test set. We compare CKA and Orthogonal Procrustes’ rank correlations (CCA metrics perform trivially since $n>p$). Averaging the results over 6 layers (every other convolutional layer omitted for computational efficiency), we see that CKA and Procrustes achieve similar rank correlations; for $\tau$, CKA has a higher correlation, and for $\rho$, Procrustes does:
> > >
> > > |Probing task |	Procrustes   $\rho$   | 	Procrustes   $\tau$   | 	 CKA $\rho$ | 	CKA $\tau$ |
> > > |-------------| -------------| -------------|-------------|-------------|
> > > | CIFAR-100 |	0.485 |	**0.376** |	**0.507**|	0.359 |
> > > | SVHN |	0.363 |	**0.272**|	**0.372** |	0.255 |
> > >
> > > For context, on the language version of this benchmark already in the paper, CKA slightly outperformed Procrustes, which both did significantly better than PWCCA. We see here that CKA and Procrustes are again close, which supports the conclusion that Procrustes is a well-rounded metric that captures functional differences across many settings.

---

> > ### Comment · Reviewer_SGnL · 2021-08-21
> > **Excellent follow-up experiments**
> >
> > I'm quite satisfied with your response. Extending your analyses to vision models was a substantial (though in my opinion necessary) request, and you did so thoroughly and with convincing results. You showed that some of my concerns were attributable to misunderstanding or misinterpretation, and I appreciate the clarification and additional details you provided. I hope that the extra page provided for revised manuscripts allows you to implement these clarifications in the text. I'm still not convinced that your enthusiasm for OPA is fully warranted by the magnitude of the results, though I agree that it seems overlooked by the literature compared to other methods, especially given its performance.
> >
> > Thank you for putting the work into your response that allows me to change my score from a 3 to 7.

---

### Official Review · Reviewer_DsGH · 2021-07-20

**Rating:** 6
**Confidence:** 3

**Summary:**

This paper evaluates recent efforts to analyze learn representations obtained by neural networks using canonical correlation analysis (CCA), centered kernel alignment (CKA), and orthogonal Procrustes distance. The authors pointed out on some inconsistencies across these metrics and suggested a method to measure of how well the distance metric is with respect to some functionality.


**Limitations And Societal Impact:**

Limitations: See above.
Societal impact: I do not see any.

**Main Review:**

The paper is clearly written, easy to follow, and the findings regarding CCA, CKA, and orthogonal Procrustes are interesting and valuable.
I'm a little bit puzzled by the quantitative measure and its contribution.
If I understand it correctly, the authors first claim that the current metrics are not consistent concerning sensitivity and specificity. The authors demonstrated it with an example. Then, the authors presented a different method, grounded by some downstream task, using the same example to assess the same thing. Can we use this method to better understand representations similarities or only to compare different distance metrics?

Overall, the method and analysis is interesting. I have some questions to the authors:
1) I agree with the authors that according to Figure 1 two different networks’ representations at layer 7 have higher PWCCA distance than that between layer 7 and any other layer within the same network. However, we still see the same behavior within different layers for different models, meaning the distance is the smallest to layer 7 than any other layer. Can the authors say something about that? Maybe this is a normalization issue? Will it be different if we do it with randomly initialized network / network that was trained for other task?
2) In line 257-259 the authors wrote: "The representations were computed on in-distribution MNLI data, meaning that the dissimilarity measures can detect OOD differences without access to OOD data". However, the authors did use OOD data (HANS) to get accuracies and learn f, no? Am I missing something?
3) The authors mentioned that they trained several models for each setting and averaged the results, can the authors also share the STDs / variance?


**Time Spent Reviewing:**

4-5 hours

---

> ### Author Response · Authors · 2021-08-10
> **Response to Reviewer DsGH**
>
> We thank the reviewer for their comments and feedback. We appreciate that the reviewer found the results to be interesting and valuable, and that the paper was clearly written. Below we discuss some of the questions the reviewer raised:
>
> >Can we use this method to better understand representations similarities or only to compare different distance metrics?
>
> We envision that our method can be used to better understand representations, and not only distance metrics. To measure representation similarity, a metric is essential; to interpret that metric, we need to know whether and when to trust it. Our benchmarks help us understand that (for instance, CKA might be under-sensitive) and so allows us to better interpret what the similarity numbers mean.
>
> Further, by assembling benchmarks of models that vary in functional behavior, our method can help uncover interesting new insights directly about representation similarity. For example, we might ask whether different functional behaviors correlate with each other, and whether optimizing for one also leads to gains on another; we could also investigate whether there is an intrinsic dimensionality to representations, and which of those dimensions differ when random initialization, architecture choice, and/or training parameters are changed.
>
> > I agree with the authors that according to Figure 1 two different networks’ representations at layer 7 have higher PWCCA distance than that between layer 7 and any other layer within the same network. However, we still see the same behavior within different layers for different models, meaning the distance is the smallest to layer 7 than any other layer. Can the authors say something about that? Maybe this is a normalization issue?
>
> There should not be a normalization issue because all representations are already centered and normalized before distances are computed (and distance metrics are already naturally on a [0,1] or [0,2] scale). For a distance metric to be useful, we think that it is not enough to see the same ordering of distances, once we fix the network itself -- we want the overall ordering of distances (including across networks) to be meaningful. In practice, we may be interested in the representational differences between networks that have partially overlapping sources of randomness (e.g. some sharing fine-tuning seed, and some sharing pre-training seed), and if distances are dominated by these factors rather than underlying functionality, we cannot trust the distances.
>
>
> >In line 257-259 the authors wrote: "The representations were computed on in-distribution MNLI data, meaning that the dissimilarity measures can detect OOD differences without access to OOD data". However, the authors did use OOD data (HANS) to get accuracies and learn f, no? Am I missing something?
>
> We use OOD data (HANS) to get accuracy scores, but not to learn $f$ (there is no learning in $f$; it is just the accuracy of the model, and the model is given). Since the question is whether IID representations can predict OOD accuracy differences, we must use OOD data to get accuracy scores -- if we used IID data, we’d just be asking whether IID representations predict IID accuracy differences, which was already tested in the preceding benchmarks.
>
> >The authors mentioned that they trained several models for each setting and averaged the results, can the authors also share the STDs / variance?
>
> We provide the full results across the averaged settings in the Appendix. For convenience, we also list an updated Table 1 with the STDs below (using results updated with more samples, so the exact numbers may differ slightly from computing the STDs using the Appendix uploaded originally):
>
> |Perturbation | Functionality | Procrustes   $\rho$   | Procrustes   $\tau$   | CKA $\rho$ | CKA $\tau$ | PWCCA $\rho$| PWCCA $\tau$|
> | ----------- | ----------- | ------------- |------------- |------------- |------------- | ------------- | ------------- |
> |  Pretraining seed, layer depth  |   QNLI probe  |$0.574 \pm 0.332$|	$0.435 \pm 0.247$	|$0.628 \pm 0.284$| $0.473 \pm 0.218$ |$0.491 \pm 0.355$| $0.368 \pm 0.258 $  |
> |  Pretraining seed, layer depth  |   SST-2 probe  |$0.624	\pm	0.296 $|	$0.474	\pm	0.231$	|$0.679	\pm	0.256$| $0.512	\pm	0.203$ |$0.556	\pm	0.324$| $0.421	\pm	0.242 $  |
> |  Pretraining seed, PC deletion  |   SST-2 probe  |$0.860	\pm	0.071$|	$0.677	\pm	0.096$ | $0.751	\pm	0.063$| $0.564	\pm	0.066$ |$0.870	\pm	0.072$| $0.690	\pm	0.100 $  |
> |  Finetuning seed  |   OOD HANS dataset  |$0.551	\pm	0.068$|	$0.398	\pm	0.058$	|$0.462	\pm	0.126$| $0.329	\pm	0.097$ |$0.568	\pm	0.071$| $0.412	\pm	0.061$  |
> |  Pretraining, finetuning seeds  |  OOD Antonymy stress test  |$0.243	\pm	0.072$|	$0.179	\pm	0.051$	|$0.227	\pm	0.104$| $0.160	\pm	0.070$ |$0.204	\pm	0.086$| $0.152	\pm	0.056 $  |
> |  Pretraining, finetuning seeds  |  OOD Numerical stress test  |$0.071	\pm	0.093$|	$0.049	\pm	0.063$	|$0.122	\pm	0.090$| $0.084	\pm	0.059$ |$0.031	\pm	0.080$| $0.023	\pm	0.056$  |

---

> > ### Comment · Reviewer_DsGH · 2021-08-15
> > **Response to Authors comment**
> >
> > I would like to thank the authors for providing additional clarification and results.
> >
> > After reading the rest of the reviews and comments, I would like to keep my score as is.

---

### Official Review · Reviewer_L7iZ · 2021-07-23

**Rating:** 8
**Confidence:** 3

**Summary:**

This paper focuses on giving a consistent meaning to (dis)similarity metrics among neural networks based on representations out of intermediate layers.

This consistent meaning is given by comparing the ordering that a dissimilarity measure gives to a "functional behavior". The functional behavior is defined as any function on the representations including but not limited to the accuracy of a given task.

The authors focus on three dissimilarity measures. First, they explain intuitively how these measures behave when there different initialization for the network. Then they give intuitions around what happens when they remove principal components of the representations. Then they show that they can arrive at similar conclusions using their rank comparison proposal with a functional behavior of the representations.

The authors then verify that the three measures can detect out-of-distribution differences when the pre-trained weights are the same but fine-tuning seed is different. However, they construct a benchmark to show that when the pre-trained weights are also different these three metrics do not show a statistically significant rank correlation, posing an open question around what metric is suitable in such a scenario.



**Limitations And Societal Impact:**

yes

**Main Review:**

What is the motivation behind using orthogonal Procrustes besides the fact that it's invariant to left orthogonal transformation? What is it capturing intuitively? Geometrically it seems to be relevant for representations that need to be compared modulo a rotation factor, but neural network representations do not necessarily get optimized to produce such rotation effect.

The first paragraph of page 5 and the footnote on page 5 together are a little confusing. Do you mean the smaller principle components are removed first or the largest ones? Or by quantifying you mean you sort them based on how much they decrease probing accuracy?

Besides these, I think the paper is novel enough and has good clarity. In terms of significance, I think it provides an intuitive framework for giving meaning to dissimilarity measures which is beneficial for future research in this field.

**Time Spent Reviewing:**

6

---

> ### Author Response · Authors · 2021-08-10
> **Response to Reviewer L7iZ**
>
> We thank the reviewer for their comments and feedback. We appreciate that the reviewer found our work to be novel, clear, and significant to the field. Below we discuss some of the questions the reviewer raised:
>
> >What is the motivation behind using orthogonal Procrustes besides the fact that it's invariant to left orthogonal transformation? What is it capturing intuitively? Geometrically it seems to be relevant for representations that need to be compared modulo a rotation factor, but neural network representations do not necessarily get optimized to produce such rotation effect.
>
> Our motivation for including orthogonal Procrustes in our evaluation was twofold: 1) it has previously been used in the literature to quantify the similarity of representations, and 2) many previous papers assume orthogonal invariance as a desideratum, and Procrustes is one of the simplest baselines that satisfies this (indeed it is often mentioned as a baseline in papers but not actually evaluated against, so we initially included it to “do our homework”, but to our surprise it performed very well)[1, 2].
>
> As to why orthogonal invariance is desirable in the first place, word2vec models provide one intuitive example. In the word2vec setting, word similarity is captured by the cosine similarity of their associated vectors, and orthogonal transformations will preserve the cosine similarity. More broadly, as discussed in Kornblith et. al, invariance to orthogonal transformation implies invariance to permutation, which is intuitively necessary given the symmetries in neural networks.
>
>
> >The first paragraph of page 5 and the footnote on page 5 together are a little confusing. Do you mean the smaller principle components are removed first or the largest ones?
>
> The smaller principal components are removed first. Thank you for helping us make this clearer; we will update the wording in the final draft.
>
>
> References:
>
> [1] S. Kornblith, M. Norouzi, H. Lee, and G. Hinton. Similarity of neural network representations revisited. In International Conference on Machine Learning, pages 3519-3529, 2019.
>
> [2] S. L. Smith, D. H. P. Turban, S. Hamblin, and N. Y. Hammerla. Ofine bilingual word vectors, orthogonal transformations and the inverted softmax, 2017.

---

### Decision · Program_Chairs · 2021-09-27

**Decision:**

Accept (Poster)

**Comment:**

This paper evaluates several representational similarity measures on a wide variety of benchmark tasks, and concludes that orthogonal Procrustes performs slightly better on average than other methods. All reviewers agreed that the empirical evaluation was novel and valuable, particularly with the additional results for vision models that the authors provided during the rebuttal period. Two reviewers noted that the suggestion that orthogonal Procrustes is superior to other methods was not entirely convincing given that different representational similarity methods seemed to perform best for different tasks. Nonetheless, all reviewers recommended acceptance.

The AC has some technical comments that may assist the authors in improving the camera-ready paper:

1. The definition of CCA in Eq. 2 isn’t sufficient to specify the value of PWCCA, since different scalings of the weights will lead to the same canonical correlations but different values of $\alpha_i$. If the components are assumed to have unit norm, then it does not seem like PWCCA is invariant to left orthogonal transformations as claimed.
2. The statement at the beginning of the appendix that the representations are normalized seems important for interpreting and reproducing the results in this work, and although L104-105 says that orthogonal Procrustes distance is not normalized between 0 and 1, because the representations are normalized, the Procrustes distance is normalized between 0 and 2.
3. The discussion suggests that orthogonal Procrustes gives better correlations than linear CKA for some tasks because, for diagonal matrices, orthogonal Procrustes reduces to a function of the squared distances between singular values whereas linear CKA reduces to the sum of the squared distances between the squared singular values. This seems possible, but this argument is not entirely convincing, because the difference in weighting of singular values is not the only difference between the methods: If all singular values are 1 and representations are equal-sized, orthogonal Procrustes distance reduces to the mean CCA correlation distance (up to a factor of 2), whereas linear CKA reduces to the mean squared CCA correlation distance. The AC suggests that the authors try manipulating (squaring/taking square roots of) the singular values of real representations before computing orthogonal Procrustes/CKA to ensure that the difference in efficacy between the methods can be explained by differences in their treatment of the singular values alone.

Finally, the biggest strength of this paper is its thorough empirical evaluation of the correlation between representational similarity and various functional properties of networks. Releasing the code for performing this evaluation would likely increase the paper's impact.